# Full-genome sequencing of dozens of new DNA viruses found in Spanish bat feces

Jaime Buigues,[1] Adrià Viñals,[2] Raquel Martínez-Recio,[1] Juan S. Monrós,[2] Rafael Sanjuán,[1,3] José M. Cuevas[1,3]

**ABSTRACT** Bats are natural hosts of multiple viruses, many of which have clear zoonotic potential. The search for emerging viruses has been aided by the implementation of metagenomic tools, which have also enabled the detection of unprecedented viral diversity. Currently, this search is mainly focused on RNA viruses, which are largely over-represented in databases. To compensate for this research bias, we analyzed fecal samples from 189 Spanish bats belonging to 22 different species using viral metagenomics. This allowed us to identify 52 complete or near-complete viral genomes belonging to the families *Adenoviridae*, *Circoviridae*, *Genomoviridae*, *Papillomaviridae*, *Parvoviridae*, *Polyomaviridae* and *Smacoviridae*. Of these, 30 could constitute new species, doubling the number of viruses currently described in Europe. These findings open the door to a more thorough analysis of bat DNA viruses and their zoonotic potential.

**IMPORTANCE** Metagenomics has become a fundamental tool to characterize the global virosphere, allowing us not only to understand the existing viral diversity and its ecological implications but also to identify new and emerging viruses. RNA viruses have a higher zoonotic potential, but this risk is also present for some DNA virus families. In our study, we analyzed the DNA fraction of fecal samples from 22 Spanish bat species, identifying 52 complete or near-complete genomes of different viral families with zoonotic potential. This doubles the number of genomes currently described in Europe. Metagenomic data often produce partial genomes that can be difficult to analyze. Our work, however, has characterized a large number of complete genomes, thus facilitating their taxonomic classification and enabling different analyses to be carried out to evaluate their zoonotic potential. For example, recombination studies are relevant since this phenomenon could play a major role in cross-species transmission.

**KEYWORDS** bat viruses, DNA viruses, metagenomics, viral emergence, viromics, zoonotic viruses

Bats are the largest mammalian order after rodents, with around 1,400 species distributed worldwide (1, 2), and play an important role as pollinators, pest controllers, seed dispersers, and reforesters (3). However, they are also a natural reservoir for a wide variety of viruses. Indeed, some bat RNA viruses are at the origin of zoonotic diseases (4, 5). Moreover, viruses may directly threaten bat populations, which can have important implications for ecosystem management (6, 7). Bat-specific features may explain their propensity to carry viruses. For example, it has been suggested that the evolution of metabolic mechanisms involved in flight capacity triggered pleiotropic effects related to pathogen immunity, thus increasing the susceptibility of bats to be asymptomatic carriers of viruses (8). Also, bats can form extremely large and densely populated colonies that tend to favor high rates of viral transmission (9). In this context, bat shelter disturbances may also increase contact with humans or domestic animals,

Address correspondence to Rafael Sanjuán, rafael.sanjuan@uv.es, or José M. Cuevas, cuevast@uv.es.

The authors declare no conflict of interest.

See the funding table on p. 18.

leading to an increased zoonotic risk. This threat has prompted the implementation of bat monitoring programs in several countries (10).

Numerous animal viruses have been discovered using metagenomics. These studies have significantly increased our knowledge of the global virosphere (11) and have enabled the identification of new and emerging viruses in various clinical and environmental samples (11, 12). As of September 2023, the bat-associated virus database (i.e., DBatVir) included over 19,000 sequences, half of which originated from Asia, followed by Africa, with European-origin samples representing less than 10%. In addition to this bias, most of the described bat viruses are RNA viruses, mainly coronaviruses, which account for more than half of the known sequences, while only 10% are DNA viruses. This over-representation of RNA viruses in databases is a consequence of their increased zoonotic potential (13), which has intensified efforts in their discovery of DNA viruses. Finally, the vast majority of viral sequences deposited in DBatVir are partial, usually from the viral polymerase or capsid genes, with full-genome sequences being the exception.

Spain hosts over 30 bat species and stands out as one of the European countries with the highest number of described bat viruses, mainly RNA viruses. Specifically, DBatVir reports 298 viral sequences from Spain belonging to families *Rhabdoviridae*, *Adenoviridae*, *Coronaviridae*, *Herpesviridae*, *Papillomaviridae*, *Filoviridae,* and *Picornaviridae*. These families include potentially zoonotic viruses such as lyssaviruses (14) and other rhabdoviruses (15), coronaviruses (16), herpesviruses (17), and a distant relative of ebolaviruses (18). By contrast, only 4 and 28 complete genomes of bat DNA viruses have been reported in Spain and Europe, respectively. To help correct this bias, we have used metagenomics to characterize the DNA virus fraction present in fecal samples from 189 bats, belonging to 22 species captured in different regions of Spain. Overall, the assembly of the viral reads obtained has enabled the recovery of 52 complete or near-complete viral genomes belonging to the families *Adenoviridae*, *Circoviridae*, *Genomoviridae*, *Papillomaviridae*, *Parvoviridae, Polyomaviridae,* and *Smacoviridae*, 30 of which represent novel DNA virus species.

## MATERIALS AND METHODS

### Study area and sample collection

Nylon mist nets and a harp trap (Austbat) were used to capture bats from different habitats that were abundant. Each captured animal was identified to species level, sexed, measured, weighed, and briefly placed in cotton bags to recover fresh fecal samples. Fecal samples were obtained from 189 bats captured in seven Spanish regions (Cantabria, Castellón, Lugo, Murcia, Salamanca, Teruel, and Valencia; Fig. 1) from May to October 2022. Of the 22 bat species 18, 3, and 1 belonged to the *Vespertilionidae, Rhinolophidae, and Molossidae* families, respectively. Samples from each individual were pooled in tubes containing 500 µL of 1× phosphate-buffered saline (PBS), kept cold initially, and then at −20°C until they were transported to the laboratory and stored at −80°C for further processing.

### Sample processing and nucleic acids extraction

A fraction of the samples from each of the 189 individuals were combined into a total of 25 pools, each containing between 1 and 15 samples from the same bat species (Table S1). Fecal samples from each pool were homogenized in a Precellys Evolution tissue homogenizer (Bertin) in 2 mL tubes with 1.4 mm ceramic beads, adding 1 vol of 1× PBS to obtain a final volume of 1.5 mL. Homogenization consisted of three cycles of 30 s at 6,500 rpm, with a 10-s pause between cycles. Homogenates were centrifuged at 20,000 × *g* for 3 min at 4°C, and the resulting supernatants were centrifuged again using the same conditions. Supernatants were then transferred to new tubes and filtered using Minisart cellulose acetate syringe filters with a 1.2-µm pore size (Sartorius). The filtrate was transferred to ultra-clean 2 mL tubes and 280 µL was collected for nucleic acid

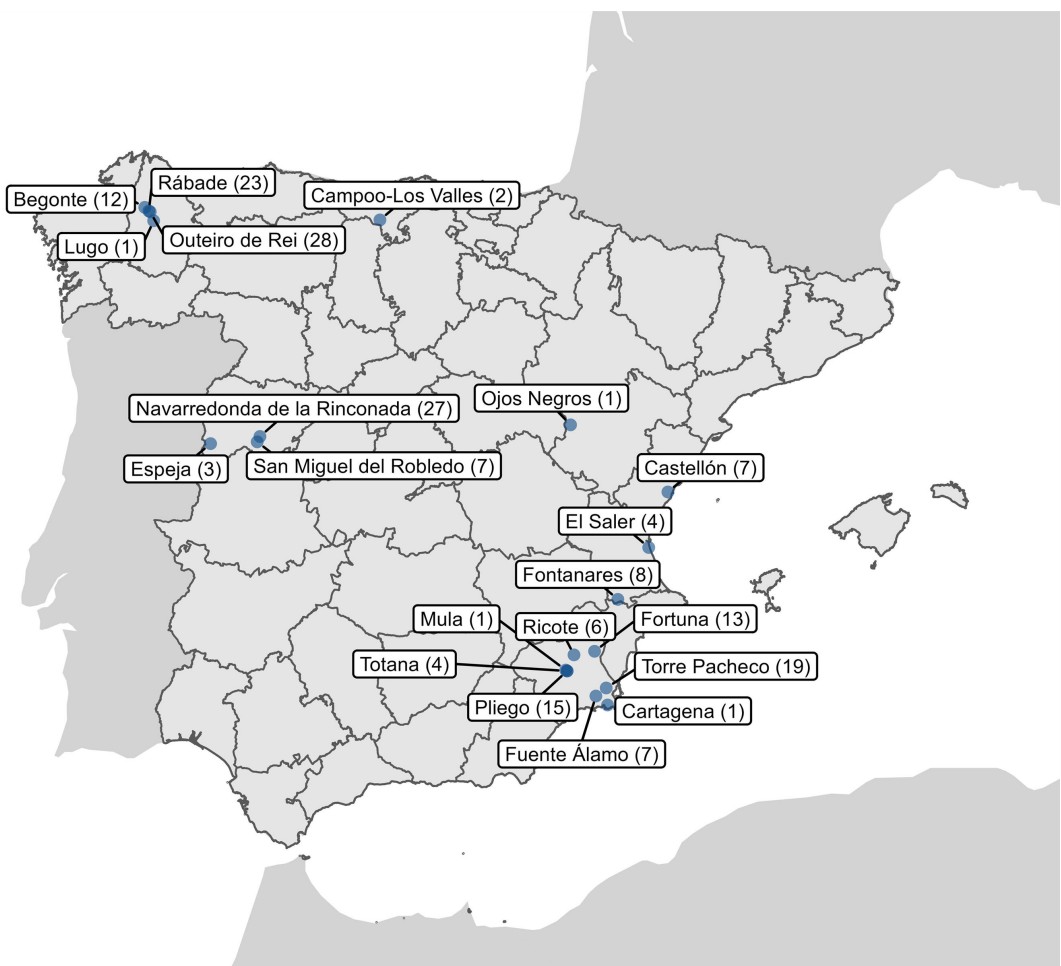

**FIG 1** Sampling points throughout Spain. The number of individuals captured in each area is indicated in parentheses. This map was created using R software (https://www.R-project.org/).

extraction using the QIAamp Viral RNA mini kit (Qiagen). The extract was eluted in a final volume of 40 µL and stored at −80°C. It is worth mentioning that, though the Qiagen kit is labeled as an RNA kit, it is also routinely used for DNA co-purification (19–23). Although only the DNA fraction has been analyzed in this study, the same samples can also be used to characterize RNA viruses.

## Sequencing and viral sequence detection

Extracted nucleic acids were used for library preparation using the Nextera XT DNA library preparation kit with 15 amplification cycles (Illumina) and subjected to paired-end sequencing in a NextSeq 550 device with a read length of 150 bp at each end. Reads were deduplicated, quality filtered with a quality trimming threshold of 20, and those reads below 70 nucleotides in length were removed using fastp v0.23.2 (24). *De novo* sequence assembly was performed using SPAdes v3.15.4 (25) with the meta option, and MEGAHIT v1.2.9 (26) using default parameters. The contigs assembled with either method were clustered to remove replicates or small replicates of larger contigs using CD-HIT v4.8.1 (27). Contigs shorter than 1,000 nucleotides were removed. The resulting clustered sequences were then taxonomically classified using Kaiju v1.9.0 (28) with the subset of NCBI nr protein database containing archaea, bacteria, and viruses, downloaded on 6 June 2023. All clustered sequences were also analyzed using Virsorter2 v2.2.4 (29) to detect viral contigs. In addition, viral contigs identified with Virsorter2 were analyzed with CheckV v1.0.1 (30) using the CheckV database v1.5 to further assess their

quality. Finally, contigs corresponding to phages and those that could not be classified into a known viral family were discarded. The remaining contigs were selected based on their size, completeness, and the ability of the assigned virus family to infect vertebrates. CheckV v1.0.1 estimates genome completeness by comparing each contig with a large database of complete viral genomes from NCBI GenBank and environmental samples (CheckV database v1.5) (30). In addition, all contigs related to the *Smacoviridae* and *Genomoviridae* families were also selected, as their ability to infect vertebrate cells has not been fully ruled out (31, 32).

## General phylogenetic analysis

Sequences similar to each contig of interest were searched using DIAMOND v2.0.15.153 (33) with the blastp option and the NCBI nr database downloaded on 7 June 2023. For each contig, the 100 closest sequences obtained from DIAMOND were retrieved and checked for association with vertebrate-infecting viruses, while protein domains were annotated using Interproscan v5.63-95.0 (34) with the Pfam database v35.0. Open reading frames (ORFs) were predicted using ORFfinder (https://www.ncbi.nlm.nih.gov/orffinder). For those sequences assigned to viruses with the potential to infect vertebrates, a multiple sequence alignment was obtained using Clustal Omega v1.2.3 (35) or MAFFT v7.490 (36), depending on whether the alignment was amino acid or nucleotide based, respectively. Phylogenetic analyses were performed using IQ-TREE v2.0.3 (37), and model selection was done using the built-in ModelFinder feature (38). Branch support was estimated with 1,000 ultra-fast bootstrapping replicates (39) and 1,000 bootstrap replicates for the SH-like approximate likelihood ratio test. Coverage statistics for the viral contigs were calculated by remapping the trimmed and filtered reads to their associated contigs using Bowtie2 v2.2.5 (40). Where indicated, pairwise sequence identities were calculated with the Sequence Demarcation Toolkit (SDT) v1.2 (41), using MAFFT for sequence alignment. In addition, viral contigs were compared to NCBI databases using BLAST (42) to obtain identity values and refine annotations.

## Family-specific phylogenetic analyses

For papillomaviruses, the Papillomavirus Episteme website (https://pave.niaid.nih.gov) (43) was initially consulted using the L1 Taxonomy Tool Analysis, which performs a pairwise alignment with the papillomavirus sequences available in this database. Then, the E1, E2, L2, and L1 nucleotide sequences from 206 representative papillomaviruses, assigned to the TaxId 151340, were downloaded from NCBI, concatenated, and aligned with MAFFT, using the GTR + F + I + G4 model to construct a maximum likelihood (ML) tree. In addition, to carry out the coevolution analysis, the associated host phylogeny was downloaded from TimeTree (www.timetree.org) (44).

For the phylogenetic analysis of viral contigs identified as polyomaviruses, large tumor antigen (LTAg) amino acid sequences were aligned with Clustal Omega, and ambiguous regions in the alignment were trimmed with trimAl v1.2rev59 (45) using the *gappyout* parameter.

For the *Parvoviridae* family, the analysis was done using the complete NS1 amino acid sequence and 126 members of the *Parvovirinae* subfamily (46). Sequences were aligned using Clustal Omega and the ML tree was computed using LG + F + I + G4 as the amino acid substitution model with 1,000 ultrafast bootstrap replicates.

For the family *Adenoviridae*, the analysis was performed using the Hexon and DNA-dependent DNA polymerase sequences. Sequences were aligned using Clustal Omega and the ML tree was computed using LG + F + I + G4 as the amino acid substitution model with 1,000 ultrafast bootstrap replicates.

For the Cressdnaviricota phylum (i.e., *Circoviridae*, *Smacoviridae*, and *Genomoviridae* families), the amino acid sequence of the replication-associated protein (i.e., Rep) was used following previous work (32, 47, 48). For *Genomoviridae* and *Smacoviridae* families, Rep alignments were also trimmed using the *gappyout* option from trimAl. In addition,

to perform genome-wide pairwise analyses, all genomic sequences were first reoriented, optimizing the position of the putative origin of replication (*ori*) using MARS (49).

## Amplification of viral sequences by PCR

For those viruses belonging to families clearly associated with mammalian hosts, PCR was used to distinguish whether each virus identified was present in a single individual sample or in several individuals from the corresponding pool, which would shed some light on the distribution of each virus in the originating bat species. Thus, viruses assigned to the *Smacoviridae* and *Genomoviridae* families were excluded from this analysis due to their likely association with foodborne transmission (31, 32). For this purpose, specific primers were designed to amplify a small region of about 500 bp for each virus of interest (Table S2). Initially, nucleic acids were extracted individually from each animal sample for the pools of interest using the QIAamp Viral RNA mini kit (Qiagen), and DNA was eluted in 30 µL. Then, 1 µL was analyzed by PCR using NZYTaq II Green Master Mix (NZYTech) and specific primers for each virus of interest on all individual samples from the pool where it was detected. To assign which samples were positive for each target virus and their geographical location, amplification products were visualized by electrophoresis using a 1% agarose gel with Green Safe Premium (NZYTech).

## RESULTS AND DISCUSSION

We obtained feces from 189 bats belonging to 22 species. These samples were processed in 25 pools, each pool containing exclusively samples belonging to the same species. Illumina sequencing from DNA samples generated between 4.9 and 23 million raw reads per pool (Table S3). At the initial filtering stage, roughly half of the reads were removed, most of which were probably duplicates generated during the PCR step included in the preparation of the sequencing libraries. To mitigate the occurrence of PCR duplicates, which can significantly increase the cost of sequencing, it is highly recommended to maximize the amount of input DNA used for library preparation (50). However, this is not possible when the amount of biological material is limiting, as was our case, thus requiring the removal of this sequencing artifact by bioinformatics methods (51).

Quality-filtered reads were *de novo* assembled, and the resulting contigs were analyzed to identify viral sequences. As a result, 35,940 viral contigs over 1 kb were obtained, of which 1,259 were complete or nearly complete. A majority fraction of these contigs were associated with bacteriophages (Table S4), as expected from metagenomic analysis in fecal samples (52, 53), and deserve further study. However, from the set of metagenome-assembled viral genomes (MAVGs), we focused on 52 MAVGs classified by Kaiju as belonging to five different families of vertebrate viruses (*Polyomaviridae*, *Papillomaviridae*, *Adenoviridae*, *Parvoviridae, and Circoviridae*), as well as two little studied families (*Smacoviridae* and *Genomoviridae*). Four of the 52 MAVGs corresponded to viruses with linear genomes belonging to *Adenoviridae* and *Parvoviridae* families (Table S5). Thirty-five of the remaining 48 MAVGs (i.e., 73%) showed terminal redundancy and could therefore be considered complete genomes. In this regard, it is important to note that genomic completeness was estimated using CheckV (30). This automatic pipeline uses an extensive customized database and performs better than alternative tools, such as viralComplete (54) and VIBRANT (55). In fact, its widespread use makes CheckV the current gold standard for genomic completeness estimation.

Each identified MAVG was remapped with reads from its sequencing library, which showed a large variability ranging from 51 reads (mean coverage depth 3.1×) assigned to MAVG33 to 2,540,468 reads (mean coverage depth 163,560×) associated with MAVG31 (Table S5). Overall, five MAVGs remapped with more than 200,000 reads (i.e., MAVG22, MAVG31, MAVG35, MAVG44, and MAVG48), thus representing a percentage ranging from 4.9% to 63% in their corresponding sequencing libraries (Tables S4 and S5). The remaining MAVGs were remapped with a percentage equal to or often much lower than 0.5% of the total number of reads. Although metagenomics studies usually detect a

multitude of reads from hosts, bacteria, or phages, it is also common to find a high representation of eukaryotic viruses (7, 56, 57). This variability in the number of reads assigned to each virus could be related to viral load. However, numerous factors strongly condition the recovery of viral reads, such as the fact that our samples consist of pools obtained from a variable number of individuals, the presence of nucleic acids from other sources, the undetermined origin of each virus (i.e., host or diet) or the potential action of inhibitory agents, such as urine, which may compromise the sequencing results. In any case, the processing of the samples in our study, which mainly involves a preliminary homogenization and filtration step, has made it possible to detect a large number of viruses, even at the whole-genome level. Therefore, our results suggest that it is not necessary to implement more laborious procedures, which may also generate additional biases, such as ultracentrifugation (58) or probe capture assays (59).

These 52 MAVGs were identified in individuals from 11 different bat species (Fig. 2). Based on Blast's analyses, nine of the MAVGs showed >85% sequence identity with previously described viruses at >90% coverage, while 43 corresponded to potential new viruses (Table S5). It should be noted that four MAVGs (i.e., MAVG3, MAVG11, MAVG36, and MAVG46) had very low coverages (i.e., <5%) with respect to the best Blast hit, which could reflect the absence of sequences with sufficient genome-wide identity in databases. It is important to note that the results of Blast and Kaiju, which is based on Blastp, merely provided a tentative taxonomic classification, which had to be further corroborated by the analyses shown below (i.e., family-specific phylogenetic reconstruction and genome annotation). Nevertheless, for each of these four MAVGs, an additional Blastp analysis was performed using the largest inferred ORF, showing a query coverage ranging from 76% to 100% and corroborating the previously assigned taxonomic classification (Table S5). Proposed names and accession numbers for newly described MAVGs are shown in Table S5. In each of the sections below, we describe the general characteristics of the identified MAVGs and discuss whether they can be considered new viral species according to the criteria established for each viral family.

## Novel members of the family *Papillomaviridae*

Four MAVGs showed a genomic organization typical of papillomaviruses: MAVG3, MAVG45, MAVG46, and MAVG49. These were detected in fecal samples from *Barbastella barbastellus* (pool P8), *Pipistrellus kuhlii* (P20), *Rhinolophus ferrumequinum* (P24), and *Plecotus austriacus* (P26), respectively (Fig. 2; Table S5). Genome sizes for all MAVGs ranged from 7.6 to 8.3 kb, the expected size for papillomavirus (60). These MAVGs encoded four early genes (E6, E7, E2, and E1) and two late genes (L2 and L1) located on the same coding strand (6) with a non-coding region between L1 and E6 genes, and also between early and late genes in MAVG46 (Table S6).

Detailed genome analysis of these four MAVGs identified several common amino acid motifs in papillomaviruses (6, 61) (Table S6). The two typical zinc-binding domains $CX_2CX_{29}CX_2C$ separated by 36 amino acids were present in the E6 protein. The same motif was also present in E7 protein, but with a small modification already described in MAVG3, MAVG45, and MAVG49 (6). The ATP-binding domain [$GX_4GK(S/T)$] was detected in the E1 protein of all MAVGs, whereas the leucine zipper domain ($LX_6LX_6LX_6L$) could not be identified in the E2 protein in any case. In addition, the retinoblastoma protein (pRB) binding motif (LXCXE) was found only in MAVG46. Regarding the presence of regulatory elements (6, 61, 62), all MAVGs showed polyadenylation signals (AATAAA), TATA boxes in the non-coding region between L1 and E6 genes, and the E1-binding sites (E1BS) [AT(A/G/T)G(C/T)(C/T)] in all non-coding regions. Finally, the E2-binding sites (E2BS) (ACC-$N_6$-GGT) were undetected only in MAVG49 (Table S6).

To ascertain the precise geographic origin of each virus, we used sequence-specific primers to test by PCR individual sample from the pools containing these four MAVGs (Tables S1, S2, and S5). This showed that MAVG3 (identified in pool P8) was present in three animals captured in Northern Spain (Begonte, Lugo), whereas MAVG49 (pool P26) was detected in a single individual captured in a nearby location (Outeiro de Rei,

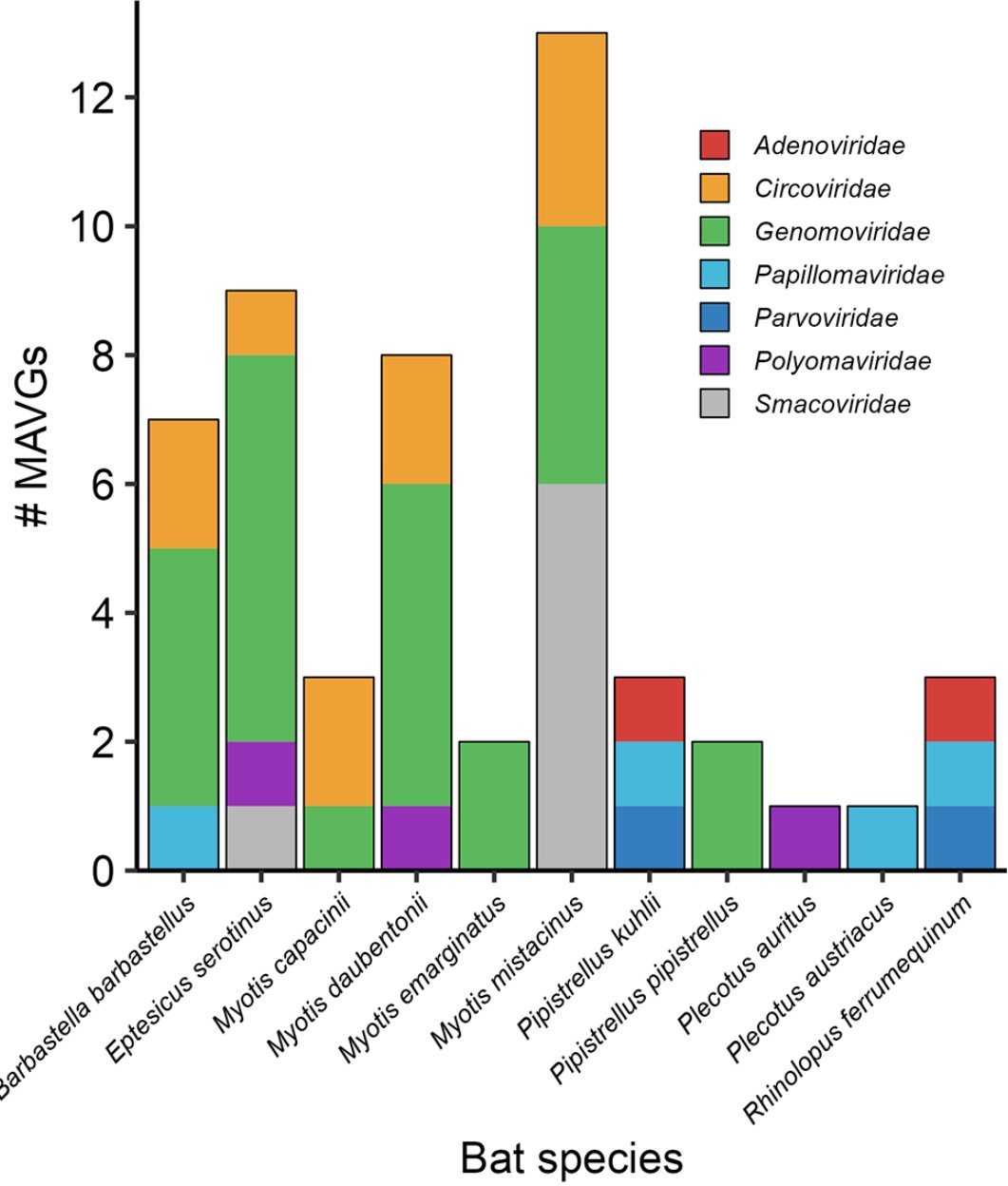

**FIG 2** Distribution of MAVGs per bat species/pool. Viral families are shown in different colors.

Lugo), and MAVG45 (pool P20) was present in three animals captured in Eastern Spain (Fontanars del Aforins, Valencia). Finally, MAVG46 was found in pool P24, which only included two animals from Fuente Álamo (Murcia), so no further analysis was required in this case. Despite the small number of animals sampled, these results suggest that at least some of the papillomaviruses identified are widely distributed in the populations tested.

The sequences of the four MAVGs initially identified as papillomaviruses were submitted to the International Animal Papillomavirus Reference Center, which validated the taxonomy and proposed a standard nomenclature (Table S5). Papillomavirus taxonomy is based on nucleotide sequence identity across the L1 gene (60). Two papillomaviruses belong to the same genus if their sequence identity in this gene is more than 60%, whereas sequences that share >70% identity are considered viral variants of the same species. MAVG3 and MAVG45 shared 71.3% and 77.4% sequence

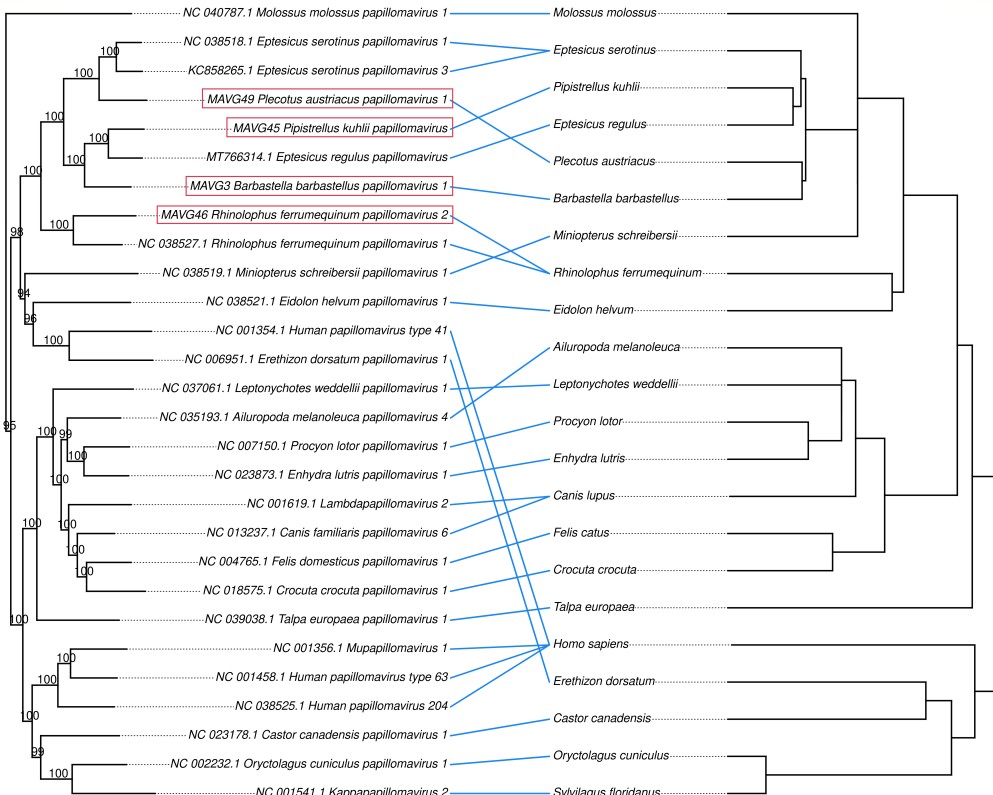

**FIG 3**  Optimized tanglegram between a papillomavirus subclade of the ML tree obtained from concatenated E1, E2, L2, and L1 nucleotide sequences (63) and associated host species. The host species tree was downloaded from www.timetree.org. The newly described viruses are highlighted in red boxes. Bootstrap values are shown at nodes. Both trees are rooted at the midpoint.

identity, respectively, with Eptesicus regulus papillomavirus (Acc. MT766314.1), an unclassified papillomavirus from Australian bats, and thus were variants of the same species. We note that both MAVGs were detected in different bat species, also distinct from the Australian variant, which reveals cross-species transmission. In addition, this is the first time that variants of the genus including this species have been described outside Australia. MAVG49 shared 73.6% sequence identity with Eptesicus serotinus papillomavirus 1 and 3 (Acc. NC_038518.1 and KC858265.1, respectively), both isolated from Spanish bats, and was therefore considered a new type of the same species. In this case, although the geographical location is common, these viral variants have been detected in different bat species, again showing cross-species transmission. This demonstrates that at least bat papillomaviruses can infect phylogenetically closely related hosts. Finally, MAVG46 was considered a new papillomavirus species, since it showed 68.9% sequence identity with the closest sequence, Rhinolophus ferrumequi-num papillomavirus 1 (Acc. NC_038527), identified in the same bat species (6).

A recent study has shown direct evidence of virus-host coevolution in a subclade including several bats and other mammalian papillomaviruses (63). Since our MAVGs were embedded in this subclade (global tree not shown), we decided to replicate this previous analysis to test whether the observed cross-species transmission events could compromise coevolution detection. To do so, the subclade of interest was selected from the global papillomavirus tree, as previously described (63), and a tanglegram including the obtained tree with the cytochrome B nucleotide sequences of the associated host species was then used (Fig. 3). The Wasserstein distance (63, 64) between the host and virus phylogenetic trees was 0.25 (two trees are topologically identical when the Wasserstein distance is 0). In addition, we used the Procrustean

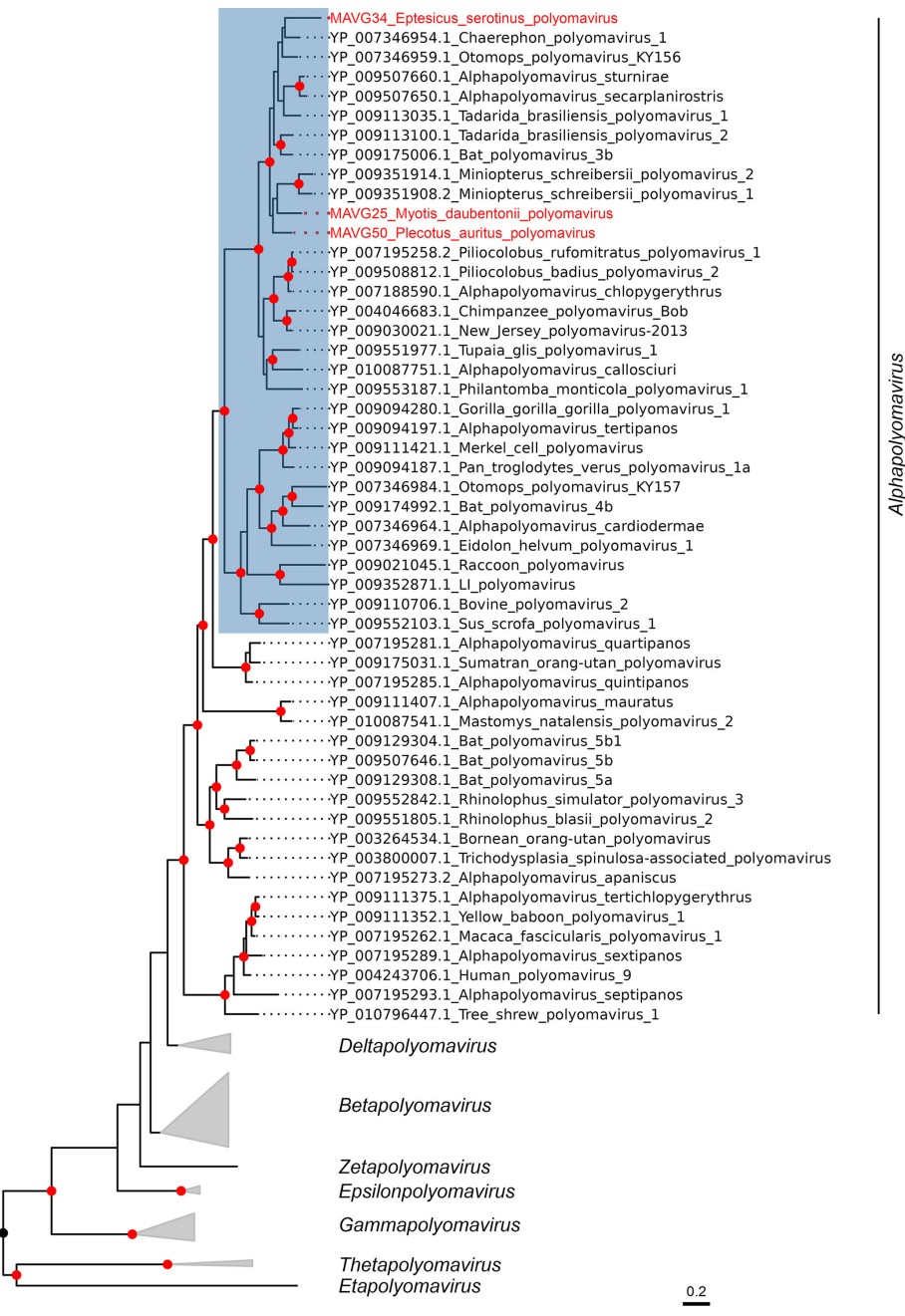

**FIG 4** ML tree of the family *Polyomaviridae* using 135 RefSeq LTAg amino acid sequences (NCBI TaxId: 151341). Taxonomic groups are collapsed by the genus. Only taxa belonging to the genus *Alphapolyomavirus* are explicitly indicated, and the group known as the VP3-less clade is highlighted in blue. Taxa are denoted by GenBank protein accession number and virus name, and novel viruses are labeled in red. Phylogenetic analysis was done using the substitution model LG + F + I + G4. SH-aLRT and bootstrap values higher than 80 and 95, respectively, are indicated with red circles. The tree is rooted at the midpoint. The scale bar indicates the evolutionary distance in amino acid substitutions per site.

Approach to Cophylogeny (PACo) (65) to assess the congruence between the viral and host phylogenies. The observed best-fit Procrustean superposition (3.78) lay outside the 95% confidence interval of the ensemble of 1,000 network randomizations in the null model. These results confirm that, at the local level, co-speciation may be a determining factor in the evolution of papillomaviruses, as previously shown (63). Globally, however, the evolutionary history of papillomaviruses is more complex, with

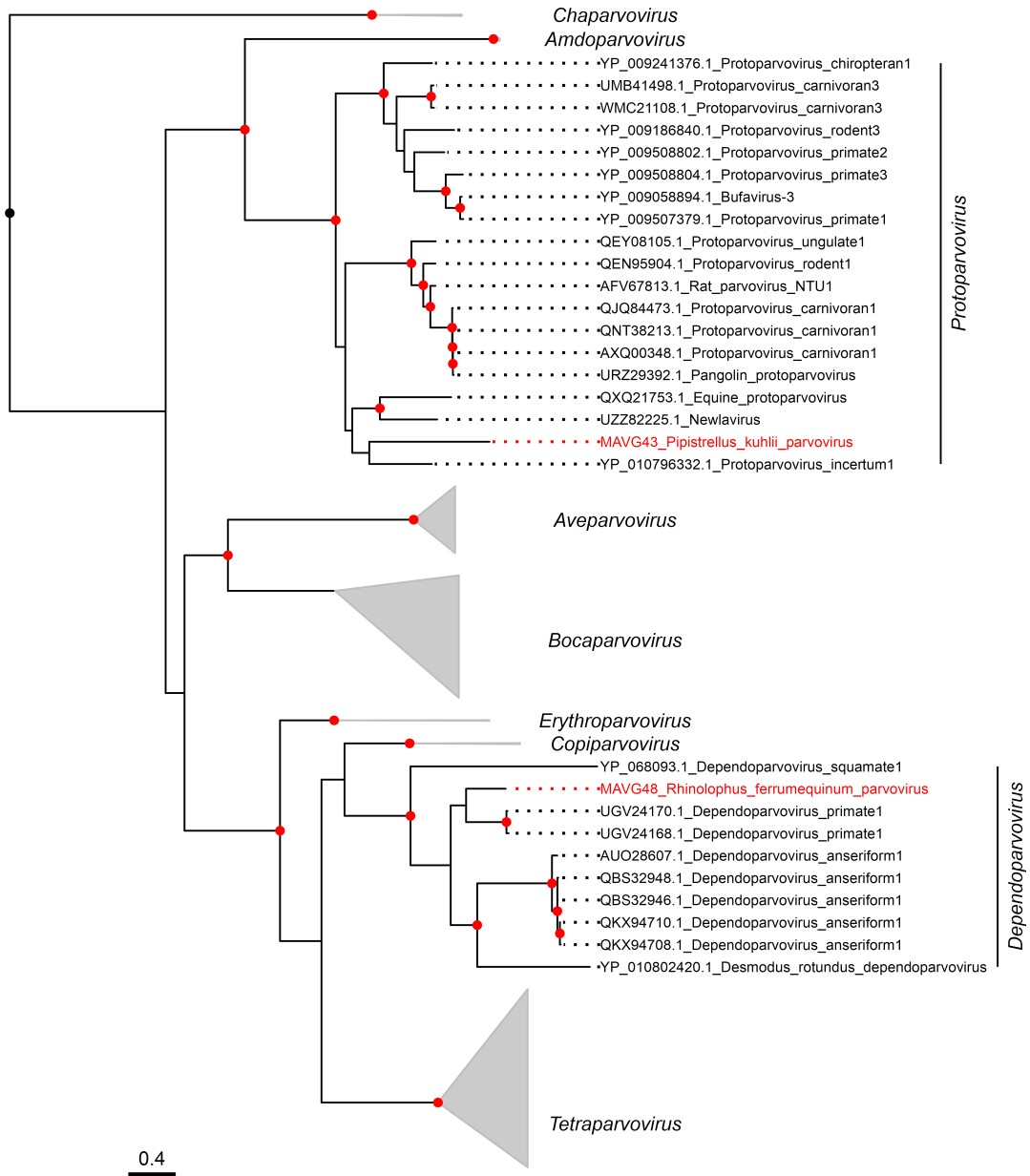

**FIG 5** ML tree of the Parvovirinae subfamily using 126 NS1 amino acid sequences. Taxonomic groups are collapsed by genus except for *Dependoparvovirus* and *Protoparvovirus* genera. Taxa are denoted by GenBank protein accession number and virus name, and novel viruses are labeled in red. Phylogenetic analysis was done using the substitution model LG + F + I + G4. SH-aLRT and bootstrap values higher than 80 and 95, respectively, are indicated with red circles. The tree is rooted at the midpoint. The scale bar indicates the evolutionary distance in amino acid substitutions per site.

multiple polyphyletic lineages infecting the same host, such as primates, rodents, or dolphins (6). For example, the clade grouping several genera of bat papillomaviruses also included human papillomavirus (human papillomavirus type 41; Fig. 3). This suggests that other evolutionary mechanisms, like intra-host divergence or niche adaptation, likely contribute to the papillomavirus phylogenetic tree (66, 67).

## Novel members of the family *Polyomaviridae*

Three polyomavirus genomes, MAVG25, MAVG34, and MAVG50, were detected in fecal samples from *Myotis daubentonii* (pool P11), *Eptesicus serotinus* (pool P14), and *Plecotus*

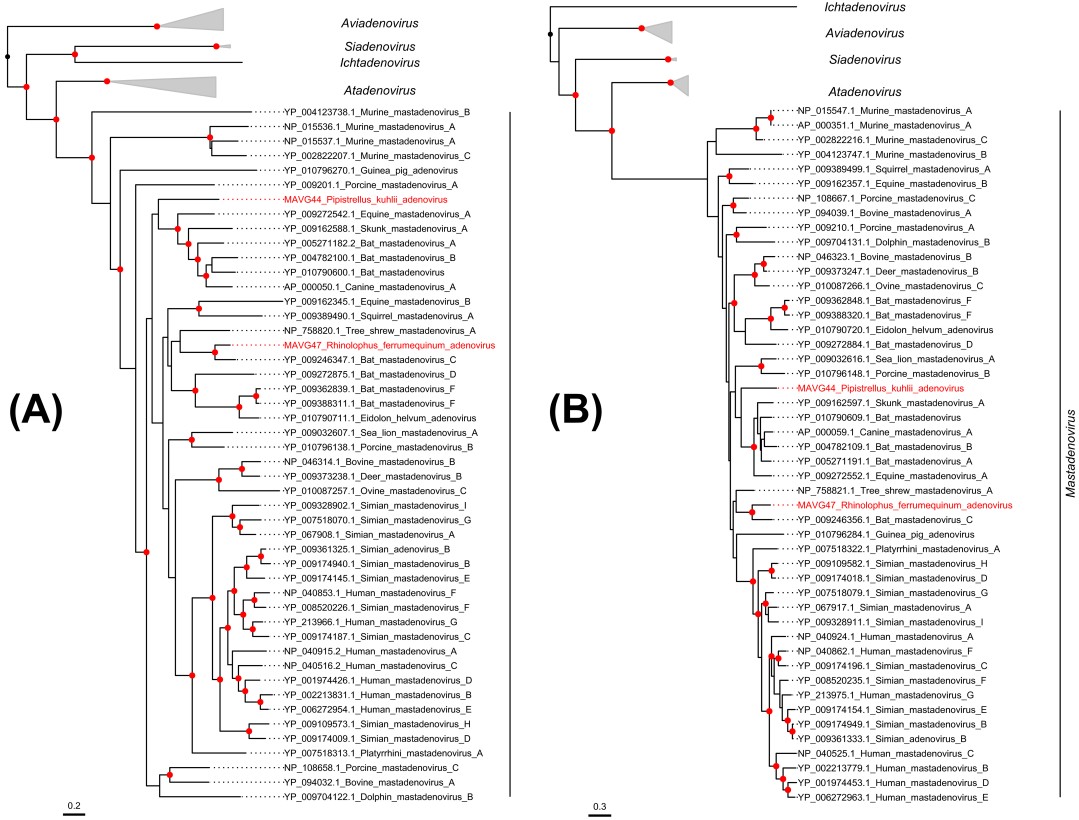

**FIG 6** ML trees of the *Adenoviridae* family using DNA polymerase (A) and hexon (B) amino acid sequences from 73 representative members. Taxonomic groups are collapsed by genus, except for the genus *Mastadenovirus*. Taxa are denoted by GenBank protein accession number and virus name, and novel viruses are labeled in red. Phylogenetic analyses were done using the substitution model LG + F + I + G4. SH-aLRT and bootstrap values higher than 80 and 95, respectively, are indicated with red circles. The tree is rooted at the midpoint. The scale bar indicates the evolutionary distance in amino acid substitutions per site.

*auritus* (pool P27), respectively (Fig. 2; Table S5). The length of the MAVGs ranged from 4.8 to 5.0 kb, in line with the expected size for a polyomavirus (68).

Polyomaviruses exhibit a conserved genome organization consisting of an early region, a late region, and a non-coding regulatory region in between (68). MAVG34 and MAVG50 showed a genome organization typical of polyomaviruses, presenting early expressed regulatory genes [encoding LTAg and small tumor antigen (STAg)] and late expressed protein genes (VP1 and VP2) (68). However, MAVG25 showed a slightly different organization, as LTAg and STAg protein domains, contrary to the usual pattern, are located in the same ORF. Also, the size of the STAg of MAVG25 is smaller than that of the other MAVGs, and that is usually described (68). Additional details on the expected conserved amino acid motifs in LTAg and STAg for the three MAVGs have also been provided (Table S7).

As above, PCR was carried out for individual samples to reveal the precise geographical location of these viruses (Tables S1, S2, and S5). This showed that all were present in very close locations in Northwestern Spain. Specifically, MAVG34 and MAVG50 were detected in two individual samples obtained from Outeiro de Rei (Lugo) pertaining to pools P14 and P27, respectively, while MAVG25 was detected in two individual samples from pool P11 obtained at another location from the same province (Rábade, Lugo; Table S1).

According to the phylogenetic analysis of the LTAg (68, 69), these three MAVGs belong to the genus *Alphapolyomavirus*, and more specifically, they are located within a monophyletic group characterized by the absence of the VP3 protein and a long VP1 (70) (Fig. 4). This group is also known as Merkel cell polyomavirus group or VP3-less

clade (70), and includes numerous viruses from bats, but also many other mammals, such as various primates, including humans. Since the three MAVGs showed less than 85% sequence identity in LTAg with other polyomaviruses, they represent new species according to ICTV criteria. Specifically, MAVG25 showed a peak sequence identity of 73.5% with Myotis davidii polyomavirus (Acc. LC426673.1), an unclassified polyomavirus isolated from *Myotis davidii* in China. MAVG34 showed a maximum sequence identity of 78.4% with an unclassified polyomavirus isolated from *Pipistrellus pipistrellus* in China (Acc. LC426677.1). Finally, MAVG50 showed the highest sequence identity (76.10%) with an unclassified bat polyomavirus isolated from *Tadarida brasiliensis* in Brazil (Acc. NC_026015.1).

Infections of different species of horseshoe bats by the same polyomavirus have been described (71), providing evidence that short-range host switching of polyomaviruses is possible in some cases. Thus, the reported new polyomaviruses are unlikely to be able to infect human cells, but their characterization may help elucidate the evolutionary history of polyomaviruses and clarify the conditions for important host-switching events.

## Novel members of the family *Parvoviridae*

Two parvovirus genomes, MAVG43 and MAVG48 were detected in fecal samples from *Pipistrellus kuhlii* (pool P19) and *Rhinolophus ferrumequinum* (P24), respectively (Fig. 2; Table S5). Both MAVGs showed the typical parvovirus genome organization, encoding the nonstructural protein 1 (NS1), and a single capsid protein (VP) (72). In addition, their genome lengths (5,351 and 4,606 nt, respectively) are within the range of the family *Parvoviridae* (72). The SF3 helicase domain was present in the NS1 of both MAVGs, whereas the phospholipase A2 domain was only detected in the VP of MAVG48 (Table S8). PCR using NS1-specific primers (Tables S1, S2, and S5) led to the detection of MAVG43 in a single sample from pool P19 collected in Eastern Spain (Fontanar dels Aforins). MAVG48 was detected in a pool containing only two individuals captured in the same location (Fuente Álamo, Murcia) and hence no PCR was done to identify the virus in individual samples.

Parvovirus species are defined using an 85% identity threshold for the NS1 amino acid sequence (46). MAVG43 belongs to the genus *Protoparvovirus* within the *Parvovirinae* subfamily (Fig. 5), but BLASTp analysis of its NS1 sequence against protoparvoviruses only showed a peak sequence identity of 44.3% and a coverage of 86% with Protoparvovirus carnivoran1 (Acc. MT815972.1). Consequently, MAVG43 is a new protoparvovirus species. MAVG48 was assigned to the genus *Dependoparvovirus* (Fig. 5), also included in the *Parvovirinae* subfamily, and showed a maximum sequence identity of 98.4% with Adeno-associated virus Croatia cul1_12 (Acc. QHY93489.1) in the NS1 protein sequence. Therefore, MAVG48 is a very close variant of a virus described in another European country, but it should be noted that this is the first time that a dependoparvovirus has been detected in Spanish bat populations.

Both *Protoparvovirus* and *Dependoparvovirus* genera contain viruses from different mammals, such as bats, rodents, and primates. Within the genus *Dependoparvovirus*, there are adeno-associated viruses that infect humans but are considered as non-pathogenic (73). Furthermore, human-associated protoparvoviruses have been detected in recent years, mostly in metagenomic fecal studies. Some of these protoparvoviruses have been found in individuals with gastrointestinal disease (74). Parvoviruses have undergone species jumps and also exhibit high levels of genome variation, similar to RNA viruses (75). The new protoparvovirus described here, Pipistrellus kuhlii parvovirus, was associated with a bat species that lives in close proximity to humans and their pets. This close contact is a risk factor for zoonotic infections, given that protoparvovirus host-switching events are believed to involve cats, dogs, and raccoons (75, 76).

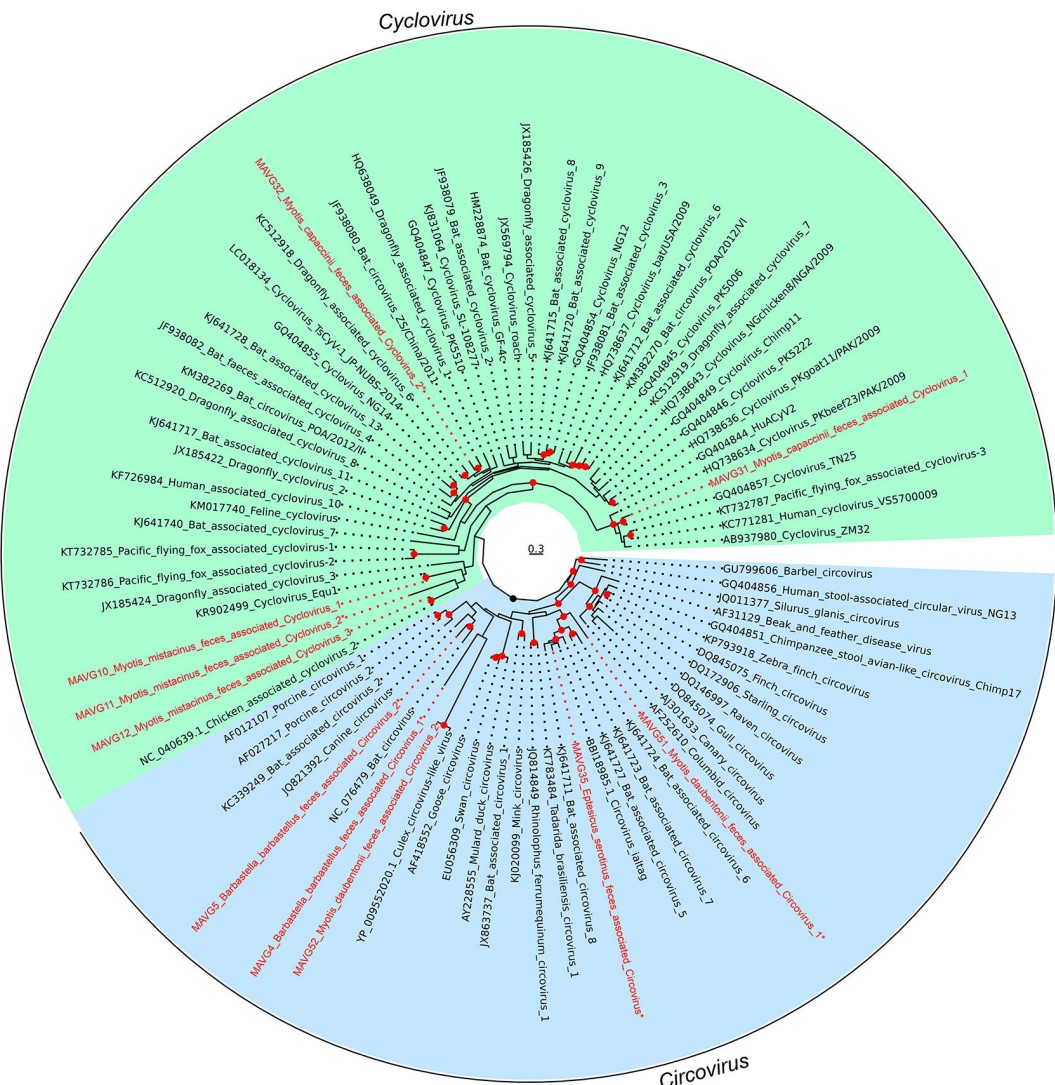

**FIG 7** ML tree of the *Circoviridae* family based on Rep amino acid sequence. Taxa are denoted by GenBank accession number and virus name, and viruses found in this study are indicated in red, while new species are indicated by an asterisk. Sequences were downloaded from the ICTV Circoviridae data resources (27 November 2023). In addition, 4 RefSeq sequences (NC_076479, NC_040639.1, BBI18985.1, and YP_009552020.1) were added to illustrate its similarity with novel MAVGs. Phylogenetic analysis was done using the substitution model LG + F + R6. SH-aLRT and bootstrap values higher than 80 and 95, respectively, are indicated with red circles. The tree is rooted to define monophyletic groups of each family genus. The scale bar indicates the evolutionary distance in amino acid substitutions per site.

## Novel members of the family *Adenoviridae*

Two adenovirus genomes, MAVG44 and MAVG47, were detected in pooled fecal samples from *Pipistrellus kuhlii* (pool P20) and *Rhinolophus ferrumequinum* (P24), respectively (Fig. 2; Table S5). Both viruses belonged to the genus *Mastadenovirus* (Fig. 6) and showed the typical genome organization of this group. MAVG44 had a GC content of 55%, in the range described for mastadenoviruses (77), and presented inverted terminal repeats (ITR) of 32 bp at both ends of the genome and 22 ORFs with putative coding sequences. As expected for the E3 region of non-primate mastadenoviruses, which is usually much simpler and shorter (78), MAVG44 showed a putative E3 region including a single ORF of 3671 nt. Although no protein domains were detected in this ORF, a BLASTp search showed 30% sequence identity and 90% coverage with the E3L protein of an Australian bat mastadenovirus (Acc. QGX41974.1). PCR analysis showed that MAVG44 was present

in three individuals from pool P20 sampled at the same location (Fontanar dels Aforins; Tables S1, S2, and S5). MAVG47 was found in a pool of two samples from the same location (Fuente Álamo, Murcia), so no PCR testing was done in this case. This genome had a GC content of 45.7% and presented ITRs of 58 bp at both ends of the genome and 22 ORFs with putative coding sequences. MAVG47 showed a single ORF for E3, which contained immunoglobulin domains (IPR007110) and exhibited 30.7% amino acid sequence identity and 97% coverage with the E3L protein of bat mastadenovirus WIV9 (Acc. YP_009246364.1).

Taxonomic classification of mastadenoviruses is usually done using a non-structural protein, such as DNA polymerase, and a structural protein (e.g., hexon protein) (79). ML trees for two different proteins showed that MAVG44 and MAVG47 clustered with non-primate adenoviruses (Fig. 6). Species definition is a complex task in mastadenoviruses, as it depends on several factors, such as phylogenetic distance, genome organization, or host range, among others. In any case, for MAVG44, a BLASTp search of the DNA polymerase and hexon amino acid sequences showed a maximum sequence identity of 76.6% and 83%, respectively, with a mastadenovirus found in *Chalinolobus gouldii*, an Australian bat (Acc. QGX41974.1). MAVG47 showed a peak sequence identity of 81.6% and 82.9% for the DNA polymerase and hexon sequences, respectively, with bat mastadenovirus WIV10 (Acc. YP_009246389.1), a member of the bat mastadenovirus C species isolated from *Rhinolophus sinicus* in China.

Adenoviruses are believed to be highly abundant in European bats (60), but additional sequencing efforts would be needed to achieve a more genome-wide characterization of these viruses. In this study, we have identified two complete genomes. However, due to the large genome size of adenoviruses, metagenomic studies typically yield partial sequences (80, 81). Previous work based on partial hexon sequences has suggested that cross-species transmission may have occurred between human and bat hosts (82). Obtaining complete genomes may help to address this more thoroughly and to identify the origins of recombination events that could play a major role in cross-species transmission (83).

## Novel members of the family *Circoviridae*

Ten circovirus MAVGs were detected in five pooled fecal samples from *Barbastella barbastellus* (P8; MAVG4 and MAVG5), *Myotis mistacinus* (P9; MAVG10, MAVG11, and MAVG12), *Myotis daubentonii* (P11; MAVG51 and MAVG52), *Myotis capaccinii* (P13; MAVG31 and MAVG32), and *Eptesicus serotinus* (P14; MAVG35; Fig. 2; Table S5). All MAVGs included two bidirectional major (>600 nt) ORFs encoding the Rep and capsid (Cp) proteins, and genome sizes ranged between 1.69 and 2.17 kb, the expected size for a circovirus genome (84). In addition, the conserved nona-nucleotide motif marking the *ori*, located between the 5′ ends of both ORFs, was detected in the intergenic region of all MAVGs, except in MAVG52, which presented the *ori* motif at the end of the Rep. PCR analysis showed that MAVG4, MAVG10, MAVG11, and MAVG12 were present in only one individual, in all cases at different locations in the province of Lugo (Tables S1, S2 and S5). MAVG51 was detected in four individuals from the province of Lugo. MAVG5 and MAVG52 were also found in four individuals each, three from Lugo and one from Salamanca. In addition, MAVG31 was present in five individuals, one from Lugo and four from Murcia. Finally, MAG32 and MAVG35 were detected in three individuals each, sampled from Murcia and Lugo, respectively.

The family *Circoviridae* includes two genera and taxa assignment to each genus is based on the location of the *ori*, which is found on the Rep or CP coding strand for the genera *Circovirus* and *Cyclovirus*, respectively. Using this criterion together with Rep phylogenetic analysis, five MAVGs were assigned to each genus (Fig. 7). Given that the species demarcation threshold is 80% genome-wide nucleotide sequence identity (84), four MAVGs (MAVG4, MAVG5, MAVG35, and MAVG51) assigned to the genus *Circovirus* could be considered new species, while MAVG52 shared a high nucleotide identity with another circovirus (Acc. PP076534) isolated from a mosquito in Belgium (Table

S5). Concerning cycloviruses, MAVG10 and MAVG12 showed a genome-wide maximum sequence identity of 92.1 and 98.9% with cycloviruses isolated from chicken feces (Acc. MN379598.1 and NC_040639.1, respectively), whereas MAVG31 presented an 82.2% with a human-associated cyclovirus (Acc. MZ201305.1).

The zoonotic potential of members of the family *Circoviridae* remains unknown. Most of the functional information available about this family comes from the study of a few members of the genus *Circovirus*, mainly porcine circoviruses (85) and beak and feather disease virus (86). In the case of the genus *Cyclovirus*, however, which has no cultured representatives, very little is known about its infectivity, transmission, or host range. Hence, the identification by metagenomics of members of this family in bat feces does not allow us to ascertain whether these are true bat viruses, particularly for cycloviruses.

## Novel members of the family *Smacoviridae*

Seven MAVGs belonging to this family were detected in two pooled fecal samples from *Myotis mistacinus* (P9; MAVG13, MAVG14, MAVG15, MAVG16, MAVG17, and MAVG18) and *Eptesicus serotinus* (P14; MAVG36; Fig. 2; Table S5). All MAVGs contained two ORFs encoding the Rep and capsid proteins in an ambisense orientation, and genome sizes ranged between 2.4 and 2.9 kb, as expected for a smacovirus genome (87). In addition, all MAVGs also showed the *ori* nonanucleotide motif described in the *Smacoviridae* family and two intergenic regions (47), except MAVG15, which only presented one. The genus demarcation threshold for this family is 40% Rep amino acid sequence identity (47). Accordingly, all MAVGs belonged to the genus *Porprismacovirus*, except MAVG36, which was assigned to the genus *Inpeasmacovirus* (Fig. 8). According to the 77% genome-wide pairwise sequence identity criterion used for delimitating species (87), all MAVGs corresponded to new species, although the high sequence identity shared by MAVG14 and MAVG16 (80.7%) grouped them as members of the same species.

The biology of smacoviruses is largely unknown, as they have not been cultured to date and have simply been associated with animals, insects, and even archaea (31, 87). Most of the members of this family have been detected in metagenomic studies of animal fecal samples (87), with a few being detected in domestic animal serum and tracheal swab samples (88, 89). Therefore, it has not been possible to assign a specific host and it is not known whether these viruses can be pathogenic for mammals or vertebrates.

## Novel members of the family *Genomoviridae*

We detected 24 MAVGs belonging to *Genomoviridae* family in seven pooled fecal samples from seven bat species (Fig. 2; Table S5). Genome sizes ranged between 2.02 and 2.36 kb, as expected for genomoviruses (90). MAVGs showed one or two ORFs encoding the Rep and one ORF encoding the capsid protein, except for MAG9, which had two. In accordance with the genus demarcation criterion, which is based on Rep amino acid sequence phylogeny, 10, 9, and 5 MAVGs were assigned to the genera *Gemycircularvirus*, *Gemykolovirus*, and *Gemykrogvirus*, respectively (Fig. 9). The species delimitation threshold is a genome-wide pairwise sequence identity of 78% (32). Accordingly, 13 new species were identified, some of which included more than one MAVG (Fig. 9; Table S9). MAVG19, MAVG21, MAVG27, and MAVG39, showed >99% genome sequence identity with other genomoviruses previously described (Table S5).

The first known genomovirus was isolated from the plant pathogenic fungus *Sclerotinia sclerotiorum* (90). Since then, more than 400 complete genomes have been described in metagenomic studies, and 10 genera have been defined (32). Members of this family have been found in insects, plants, fungi, and vertebrates (including humans), and the true extent of their host range remains unknown, as does their involvement in a pathogenic role.

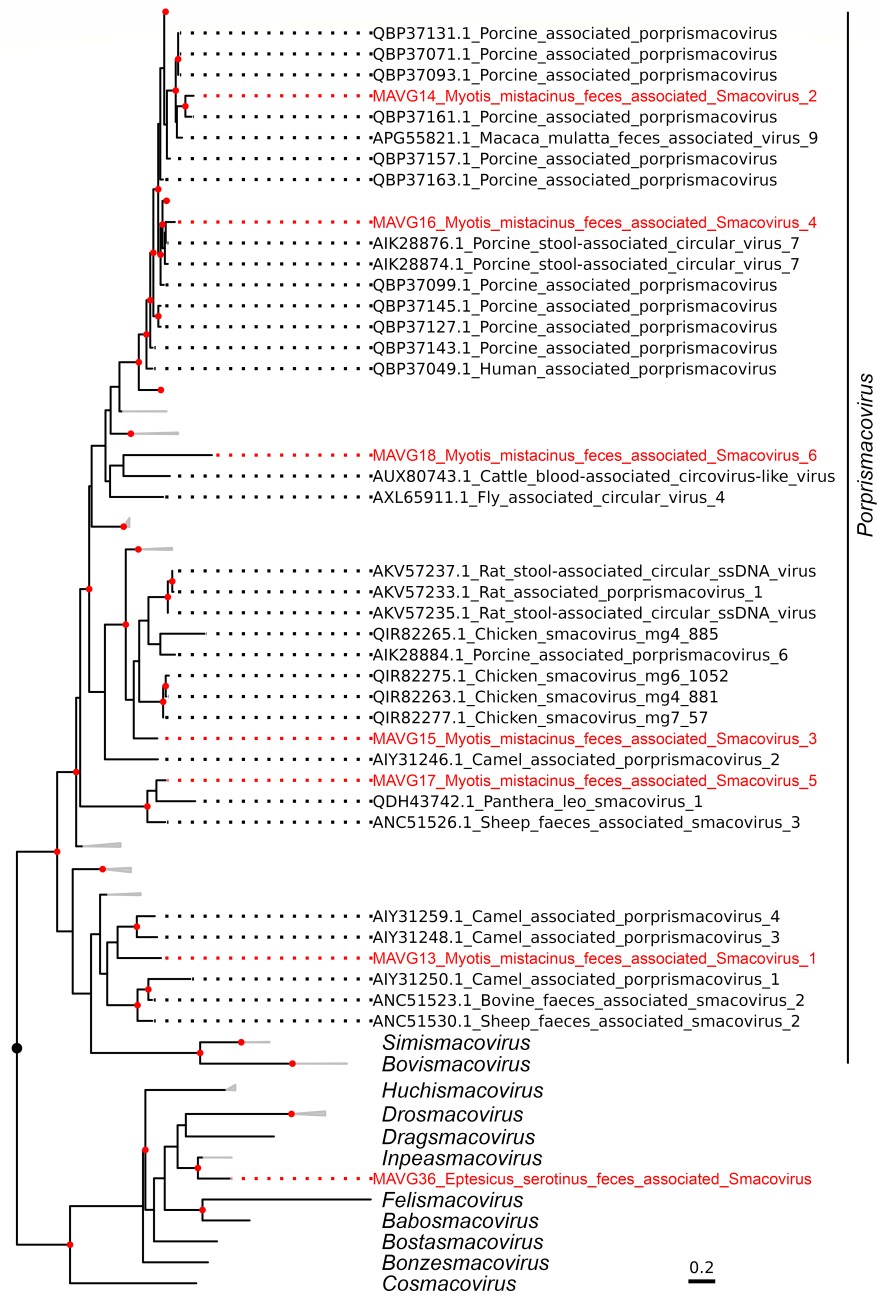

**FIG 8** ML tree of the family *Smacoviridae* based on 215 Rep amino acid sequences. Taxonomic groups are collapsed by genus, except for *Porprismacovirus* genus, and some non-illustrative clades within this genus. Taxa are denoted by GenBank accession number and virus name, and novel viruses are labeled in red. Phylogenetic analysis was done using the substitution model LG + F + I + G4. SH-aLRT and bootstrap values higher than 80 and 95, respectively, are indicated with red circles. The tree is rooted at the midpoint. The scale bar indicates the evolutionary distance in amino acid substitutions per site.

## Conclusions

The starting point for the study of viral emergence is the characterization of wildlife diversity. This has prioritized tropical regions, where land-use alterations, high wildlife diversity, and bush meat consumption are believed to increase disease emergence risk (91). It should be noted, though, that some emerging viral diseases have not originated in tropical areas (92, 93). Wildlife biodiversity is lower in Europe, which implies a lower zoonotic potential, but also suffers from a serious problem of destruction and

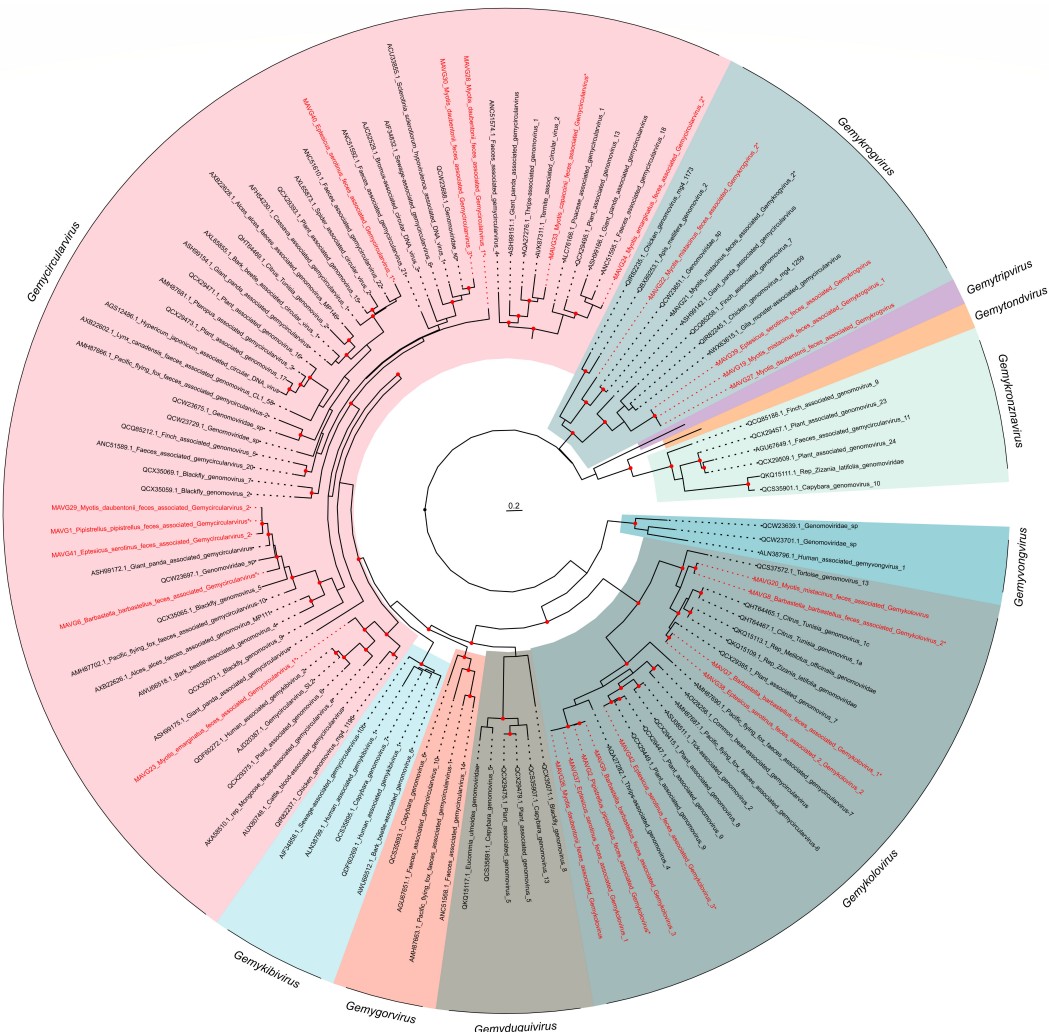

**FIG 9** ML tree of the family *Genomoviridae* based on 94 representative amino acid sequences of Rep gene. Taxonomic groups are collapsed by genus, except for those genera where new viruses are identified. Taxa are denoted by GenBank accession number and virus name, and novel viruses are labeled in red, indicating with an asterisk those that are defined as new species. When a new species includes more than one novel MAVG (see Table S9), only one is indicated with an asterisk. Phylogenetic analysis was done using the substitution model LG + F + I + G4. SH-aLRT and bootstrap values higher than 80 and 95, respectively, are indicated with red circles. The tree is rooted at the midpoint. The scale bar indicates the evolutionary distance in amino acid substitutions per site.

transformation of different habitats, particularly in Spain, which promotes closer contact between humans and wild mammals. Indeed, animal-to-human viral transmission events regularly occur in Europe (94). It is therefore necessary to undertake studies to characterize wildlife diversity in this region and to develop local viral surveillance programs, which will improve our ability to respond to potential outbreaks. For this purpose, bats, due to their high potential to harbor zoonotic viruses, are the primary action target.

RNA viruses have a higher zoonotic potential than DNA viruses (95). However, this risk is also present for some DNA virus families, where there are examples of viruses with zoonotic potential which represent a threat to both animal populations and public health. Our study design was not intended to draw epidemiological conclusions, but primarily to reflect the existing diversity of DNA viruses in Spanish bats. In particular, it is worth emphasizing here that our work was not intended to show a global view of the viral diversity present in the samples analyzed, but focused on the characterization of complete or near-complete viral genomes. It is therefore not surprising that most of the MAVGs identified correspond to viruses of small genome size, with the exception of adenoviruses. Other viruses of larger size, but whose genomes were only partially

represented, would have been neglected throughout the successive stages of bioinformatic filtering. On another note, an interesting observation is that, despite the small sample size in terms of the number of animals sampled, half of the viruses analyzed by PCR were present in more than one individual, suggesting that these infections were not exceptional but could be characterized by high population prevalence. In addition, our results also point to the need to study DNA viruses to better understand key aspects, such as transmission dynamics or host range. This will allow us to discern their true zoonotic potential and to establish surveillance strategies, as is currently being considered for RNA viruses.

## ACKNOWLEDGMENTS

We would like to thank the collaboration of different research groups that have supported the fieldwork, especially Xosé Pardavila and the Morcegos de Galicia.

This research was financially supported by grant PID2020-118602RB-I00 from the Spanish Ministerio de Ciencia e Innovación (MICINN) and co-financed by FEDER funds, and grant CIAICO/2022/110 from the Conselleria de Educación, Universidades y Empleo (Generalitat Valenciana).

## AUTHOR AFFILIATIONS

[1]Institute for Integrative Systems Biology (I2SysBio), Universitat de València and Consejo Superior de Investigaciones Científicas, València, Spain
[2]Institut Cavanilles de Biodiversitat i Biologia Evolutiva, Universitat de València, València, Spain
[3]Department of Genetics, Universitat de València, València, Spain

## AUTHOR ORCIDs

Jaime Buigues http://orcid.org/0000-0002-9016-4628
Rafael Sanjuán http://orcid.org/0000-0002-1844-545X
José M. Cuevas http://orcid.org/0000-0003-2049-3554

## FUNDING

| Funder | Grant(s) | Author(s) |
|---|---|---|
| Ministerio de Ciencia e Innovación (MCIN) | PID2020-118602RB-I00 | Rafael Sanjuán |
| | | José M. Cuevas |
| Generalitat Valenciana (GVA) | CIAICO/2022/110 | Rafael Sanjuán |

## AUTHOR CONTRIBUTIONS

Jaime Buigues, Data curation, Formal analysis, Investigation, Methodology, Validation, Visualization, Writing – original draft, Writing – review and editing | Adrià Viñals, Data curation, Investigation, Methodology, Writing – review and editing | Raquel Martínez-Recio, Data curation, Investigation, Methodology | Juan S. Monrós, Conceptualization, Funding acquisition, Investigation, Methodology, Supervision, Writing – review and editing | Rafael Sanjuán, Conceptualization, Data curation, Formal analysis, Funding acquisition, Investigation, Methodology, Project administration, Resources, Supervision, Validation, Visualization, Writing – original draft, Writing – review and editing | José M. Cuevas, Conceptualization, Data curation, Formal analysis, Funding acquisition, Investigation, Methodology, Project administration, Resources, Supervision, Validation, Visualization, Writing – original draft, Writing – review and editing

## DATA AVAILABILITY

The raw sequence reads were deposited in the Sequence Read Archive of GenBank under accession numbers SRR27912327-51. The MAVGs described in this study, which corresponded to complete or near-complete genomes, were deposited Bioproject accession number PRJNA1074704 (Supplementary Table S5).

## ETHICS APPROVAL

Samples consisted of feces from wild animals captured using nylon mist nets or a harp trap. Bats were kept briefly in cotton bags until fresh fecal samples were obtained. According to the European directive regulating the protection of animals used for scientific purposes (2010/62/EU, Article 1), subsequently transposed into Spanish legislation (Royal decree 53/213, 1 February, Article 2), procedures used in this study (i.e., capture, non-invasive handling and *in situ* release of wild animals) are not subject to the condition of animal experimentation and, therefore, an IACUC approval document is not required, but specifically a permit from the competent regional authority [Ref. Exp. 2022-VS (FAU22_009)].

## ADDITIONAL FILES

The following material is available online.

### Supplemental Material

**Table S1 (Spectrum00675-24-s0001.xlsx).** Bat species, collection sites and pool distributions.
**Table S2 (Spectrum00675-24-s0002.xlsx).** Primers used for viral sample confirmation by PCR.
**Table S3 (Spectrum00675-24-s0003.xlsx).** Illumina reads and number of viral contigs obtained.
**Table S4 (Spectrum00675-24-s0004.xlsx).** List of viral contigs classified by CheckV as high quality or complete genome.
**Table S5 (Spectrum00675-24-s0005.xlsx).** Descriptions, accession numbers, and proposed names for the novel MAVGs.
**Table S6 (Spectrum00675-24-s0006.xlsx).** Genome information for papillomavirus MAVGs.
**Table S7 (Spectrum00675-24-s0007.xlsx).** Genome information for polyomavirus MAVGs.
**Table S8 (Spectrum00675-24-s0008.xlsx).** Genome information for parvovirus MAVGs.
**Table S9 (Spectrum00675-24-s0009.xlsx).** Novel genomovirus species clusters.

### Open Peer Review

**PEER REVIEW HISTORY (review-history.pdf).** An accounting of the reviewer comments and feedback.

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
