## [Reviewer comments · Microbiology Spectrum]

Microbiology Spectrum

Full genome sequencing of dozens of new DNA viruses found in Spanish bat faeces

Jaime Buigues Bisquert, Adrià Viñals, Raquel Martínez-Recio, Juan Monrós, Rafael Sanjuán, and José Cuevas

Corresponding Author(s): José Cuevas, Institute for Integrative Systems Biology, I2SysBio, Universitat de València-CSIC

Review Timeline:

Submission Date:	March 13, 2024
Editorial Decision:	April 16, 2024
Revision Received:	May 8, 2024
Editorial Decision:	June 15, 2024
Revision Received:	June 21, 2024
Accepted:	June 26, 2024

Editor: Biao He

Reviewer(s): Disclosure of reviewer identity is with reference to reviewer comments included in decision letter(s). The following individuals involved in review of your submission have agreed to reveal their identity: Ignacio G. Bravo (Reviewer #1); Kanchan Bhardwaj (Reviewer #3)

Transaction Report:

DOI: <https://doi.org/10.1128/spectrum.00675-24>

Re: Spectrum00675-24 (Full genome sequencing of dozens of new DNA viruses found in Spanish bat faeces)

Dear Dr. Jose M Cuevas:

Thank you for the privilege of reviewing your work. Below you will find my comments, instructions from the Spectrum editorial office, and the reviewer comments.

The manuscript reports DNA virome profiling of fecal samples collected from 189 Spanish bats and phylogenetic characterization of 50 complete or near-complete genomes from seven viral families. As you can see below, the reviewers raised some concerns about the manuscript, particularly the completeness of these genomes and ethical permission; thus, I invite you to revise the manuscript. I encourage you to provide a detailed description of how you determine the completeness of these genomes. Particularly, per ASM policy on animal use (<https://journals.asm.org/animal-use>), please

1. add a section on "Ethics Approvals" in the revised manuscript where you can provide the relevant IACUC approval number(s) and
2. provide the documentation itself to Spectrum staff in "manuscript files not for publication." This IACUC approval document(s) should be translated into English for our reference and review, including portions in this document about the PPE used while catching bats.

Revision Guidelines

Sincerely,
Biao He
Editor
Microbiology Spectrum

Reviewer #1 (Public repository details (Required)):

Data deposited as Bioproject PRJNA1074704, but unable to revise them because not available yet.

Reviewer #1 (Comments for the Author):

In their manuscript, Buigues and coworkers present the results of a metagenomic approach on bat faeces from different species recovered in a number of locations in Spain. The authors communicate an important number of DNA viral genome sequences potentially belonging into seven viral families.

The research question is timely, the samples are pertinent to address it and the methods are appropriate. The results are substantial in terms of quantity, and they seem to have been deposited for free access upon publication (as far as I can judge from the information in table S4). Overall, the study contributes substantially to our knowledge on DNA viruses diversity in bats in Europe. The main caveat is that it is difficult to evaluate from the presented data the appropriate completeness of the genomes communicated (a claim that is upfront presented in the title).

Specific points that need to be explicitly addressed

-Table S3.

1) roughly half of the reads could not be used, even if quality filters were not very stringent. May be worth to discuss.
2) from metagenomic sequencing projects such as the one described here, one would naively expect that the largest fraction of DNA recovered should belong to the host, to bacteria and to bacterial viruses. It seems that the amount of recovered viral genetic material probably infecting the host is very large, and this may need to be compared with similar approaches and discussed.

-Table S4.

1) is there any information about viral genome loads that could be derived from the large differences in number of viral reads recovered for the different viral contigs?
2) from the values of number of mapped reads, mean coverage and coverage standard deviation, it may very well be the case that some positions have not been covered or that have been covered with less than 10 reads and/or with non-balanced coverage between forward and reverse reads. I think that the coverage quality should specify this fact (e.g. number or percentage of positions that are covered less than 10 times), so that the reader can evaluate the completeness of the claim for complete genome sequencing.
3) some of the query cover values are very low (e.g. 1% for *Barbastella barbastellus* papillomavirus 1, or 2% for *Eptesicus serotinus* feces associated Smacovirus). This very low values demand an explanation about the interpretation of the nature of the sequences.
4) an explanation is required for the differential use of "faeces associated" in the names proposed for the viruses

-A justification for the choice of RNA extraction for DNA viral identification is probably required.

-for circular genomes, please describe how completeness was assessed.

-for linear genomes, please describe how the completeness of the sequencing in the ends of the molecule was assessed.

-maybe the phylogenetic representations could be standardised for all families, for the sake of uniformity (and elegance).

-please include the ethical statements for wildlife sampling that may be appropriate in the country.

Minor points:

P4L18: please clarify "Homogenates were centrifuged in two rounds"

P7L46: I have not checked extensively the correspondence of each reference to the claim reported in the text, but ref 50 is not appropriate here. Please verify all other statements.

Dated and signed:

Montpellier, 11/04/2024

Ignacio G. Bravo

Reviewer #2 (Public repository details (Required)):

They've already submitted their data to GenBank and SRA but not released yet.

Reviewer #3 (Public repository details (Required)):

The raw data and MAVGs are deposited in Genbank (Sequence Read Archive of GenBank under accession numbers SRR27912327-51 and the complete or nearly complete genomes, under accession numbers PP410048-97).

Reviewer #3 (Comments for the Author):

In this study, authors have analyzed DNA viruses, through metagenomics of fecal samples collected from 189 Spanish bats belonging to 22 different species. They report:

- (i) Recovery of 50 complete or near-complete viral genomes, which belong to 7 families of DNA viruses.
- (ii) 28 of the identified genomes are new viral species.
- (iii) The raw data and MAVGs are deposited in Genbank (Sequence Read Archive of GenBank under accession numbers SRR27912327-51 and the complete or nearly complete genomes, under accession numbers PP410048-97).

Strengths of the study:

1. Context of the study is presented very well.
2. Relevant tools and methods are used for data analysis.

Weaknesses of the study:

Gut microbiome in bats includes bacteria and therefore bacteriophages can be expected in addition to the eukaryotic viruses. In this report, authors found viruses that belong to various families of eukaryotic virus families only. It would be good to include this point in the discussion because in other organisms the identified viruses in the gut are mostly bacteriophages.

As a reviewer, it's essential to appreciate the diligent efforts of authors in this study. The study highlights bats as reservoirs for numerous viruses with potential zoonotic transmission. Metagenomic tools have been pivotal in uncovering unprecedented viral diversity. While current research predominantly focuses on RNA viruses, authors addressed this bias by analyzing fecal samples from 189 Spanish bats across 22 species using viral metagenomics targeted to DNA viruses. This approach led to the identification of 50 complete or near-complete viral genomes from various families, potentially including 28 new species. The fact that the collected samples are solely focused on DNA viruses is regrettable, given the richness of the bat collection, yet it is entirely at the discretion of the authors. The publication's focus appears somewhat mixed. If it is intended as a discovery study, the authors should provide more detailed genomic explanations of the newly discovered viruses. Alternatively, if the focus is on viral distribution in bats, the newly identified viruses should be screened within their collection. In the conclusion section, the authors have noted that this paper is not intended as an epidemiological study. Consequently, specific enhancements are needed, especially in the Results section. Authors are asserting the discovery of novel viruses, and to support this claim, they should include viral motifs, not only for specific genomes but also detail the similarity of other genomes to their closest viruses for each individual novel virus.

Page 5

line 29 Please, providing a separate paragraph to explain the construction of the family phylogenetic tree can enhance readability for readers.

Page 6

Line 2. Could you clarify the statement "Those viruses identified as having a high probability of infecting bats, and not originating from other sources such as diet, were analyzed by PCR"? It seems unclear how PCR was used to determine if a virus is not foodborne. Perhaps this part of the paragraph should be revised. It appears to imply that PCR was conducted to ascertain whether parvoviruses or polyomaviruses are not transmitted through food. However, I believe the intention might be to indicate that the Smacoviridae and Genomoviridae families were excluded from the analysis panel due to their association with foodborne transmission.

Line 6-13 Did you only examine individual samples from positive pools? If your answer yes, why didn't you screen all individual samples for novel viruses, especially considering their potential impact on geographical distribution? Alternatively, did you observe any similarity between the viral genomes of your novel findings and those found in contigs under 1kb?"

Line 20 What criteria guided your selection of 50 out of 1053 nearly complete genomes? I suggest including a supplementary table in the publication containing the details of these 1003 complete genomes, which would enhance the value of the publication. Additionally, you have the option to incorporate a chart illustrating the distribution of viral reads into the paper.

Line 31. None of the data submitted to Genbank or SRA has been released yet, please do not forget to release it upon publication.

Line 32. Kindly include the lengths of all novel viruses in parentheses within the paper.. E.g. MAVG3 (7674nt)

Page7

Line 10 "Could you please clarify whether the sequence identity section pertains to the entire genome or is specifically based on the L1 region?"

Line 25 You are claiming novel papillomavirus because of this could you consider including in the paper the genome similarities of each genome individually to its closest virus?"

Page 8

Line 16 The section discussing novel viruses, particularly polyomaviruses, requires improvement. Please enhance this section by detailing the conserved domains and viral motifs contained within your novel polyomavirus LT antigen.

Line 23 Please add reference.

Page 9

Line 4 Please provide a detailed information about your novel parvovirus such as length of NS1, VP, ORFs, motifs etc.

In this study, authors have analyzed DNA viruses, through metagenomics of fecal samples collected from 189 Spanish bats belonging to 22 different species. They report:

- (i) Recovery of 50 complete or near-complete viral genomes, which belong to 7 families of DNA viruses.
- (ii) 28 of the identified genomes are new viral species.
- (iii) The raw data and MAVGs are deposited in Genbank (Sequence Read Archive of GenBank under accession numbers SRR27912327-51 and the complete or nearly complete genomes, under accession numbers PP410048-97).

Strengths of the study:

1. Context of the study is presented very well.
2. Relevant tools and methods are used for data analysis.

Weaknesses of the study:

Gut microbiome in bats includes bacteria and therefore bacteriophages can be expected in addition to the eukaryotic viruses. In this report, authors found viruses that belong to various families of eukaryotic virus families only. It would be good to include this point in the discussion because in other organisms the identified viruses in the gut are mostly bacteriophages.

Response to the Editor

Thank you for the privilege of reviewing your work. Below you will find my comments, instructions from the Spectrum editorial office, and the reviewer comments.

The manuscript reports DNA virome profiling of fecal samples collected from 189 Spanish bats and phylogenetic characterization of 50 complete or near-complete genomes from seven viral families. As you can see below, the reviewers raised some concerns about the manuscript, particularly the completeness of these genomes and ethical permission; thus, I invite you to revise the manuscript. I encourage you to provide a detailed description of how you determine the completeness of these genomes.

The description of the procedure to infer the completeness of the reported genomes is now detailed in the Methods section.

Particularly, per ASM policy on animal use (<https://journals.asm.org/animal-use>), please

- 1. add a section on "Ethics Approvals" in the revised manuscript where you can provide the relevant IACUC approval number(s) and*
- 2. provide the documentation itself to Spectrum staff in "manuscript files not for publication." This IACUC approval document(s) should be translated into English for our reference and review, including portions in this document about the PPE used while catching bats.*

According to Spanish legislation, the capture, non-invasive handling and in situ release of wild animals does not require an IACUC approval document. Only a permit from the competent regional authority is required. This is now indicated in the added "Ethics approvals" section, in addition to mentioning the reference to state legislation. This document is attached as "manuscript files not for publication".

Response to Reviewer #1

In their manuscript, Buigues and coworkers present the results of a metagenomic approach on bat faeces from different species recovered in a number of locations in Spain. The authors communicate an important number of DNA viral genome sequences potentially belonging into seven viral families.

The research question is timely, the samples are pertinent to address it and the methods are appropriate. The results are substantial in terms of quantity, and they seem to have been deposited for free access upon publication (as far as I can judge from the information in table S4). Overall, the study contributes substantially to our knowledge on DNA viruses diversity in bats in Europe. The main caveat is that it is difficult to evaluate from the presented data the appropriate completeness of the genomes communicated (a claim that is upfront presented in the title).

Specific points that need to be explicitly addressed

-Table S3.

- 1) roughly half of the reads could not be used, even if quality filters were not very stringent. May be worth to discuss.*

Most of the deleted reads are deduplicates, which were most likely generated during the PCR step for the preparation of the sequencing libraries. This is now indicated in the Results and Discussion section, including a reference that specifically discusses the problem of PCR duplicates (Rochette et al., 2023, PMID 37062860), one of whose

conclusions is that it is highly recommended to maximise the amount of input DNA used for library preparation. Unfortunately, this was not feasible in our study.

2) from metagenomic sequencing projects such as the one described here, one would naively expect that the largest fraction of DNA recovered should belong to the host, to bacteria and to bacterial viruses. It seems that the amount of recovered viral genetic material probably infecting the host is very large, and this may need to be compared with similar approaches and discussed.

The processing of the samples in our study is that usually performed in metagenomics studies using faecal samples, which basically involves a preliminary homogenization stage followed by a filtration step. Following reviewer's recommendation, we have addressed this in the Results and Discussion section to emphasise that this procedure is sufficient to detect a large number of viruses, even at the whole genome level, without the need to use more laborious procedures that may generate additional biases, such as ultracentrifugation or capture assays with probes.

-Table S4.

1) is there any information about viral genome loads that could be derived from the large differences in number of viral reads recovered for the different viral contigs?

This comment is closely related to the above. We agree with the reviewer that the number of viral reads detected can obviously be related to the viral load. However, we consider that this parameter cannot be estimated in practice, because many other factors can strongly condition the recovery of viral reads, such as the fact that our samples consist of pools obtained from a variable number of individuals, the presence of nucleic acids from other sources, the undetermined origin of each virus (i.e. host, diet, or environment) or the action of inhibitory agents, such as urine, which can compromise the final sequencing results. All these factors may condition the sequencing results and, therefore, cause a bias difficult to estimate that would discourage the quantitative interpretation of the reads obtained from viral origin. Nevertheless, we appreciate the reviewer's comment and, in fact, we have included a description of the viral read ranges (and coverage) at the beginning of the Results and Discussion section, emphasising, for the reasons mentioned above, that it is inappropriate to interpret them as an approximation of viral loads for different viruses.

2) from the values of number of mapped reads, mean coverage and coverage standard deviation, it may very well be the case that some positions have not been covered or that have been covered with less than 10 reads and/or with non-balanced coverage between forward and reverse reads. I think that the coverage quality should specify this fact (e.g. number or percentage of positions that are covered less than 10 times), so that the reader can evaluate the completeness of the claim for complete genome sequencing.

Following the reviewer's recommendation, now in supplementary table (currently renamed S5), in addition to the values for mapped reads, mean coverage and deviation, two columns indicating the percentage of positions that are covered less than 10-fold (and five-fold) for each viral genome have also been added.

*3) some of the query cover values are very low (e.g. 1% for *Barbastella barbastellus papillomavirus 1*, or 2% for *Eptesicus serotinus feces associated Smacovirus*). This very low values demand an explanation about the interpretation of the nature of the sequences.*

Blast results showed very low cover values in the best hit of a small number of genomes, but this was a consequence of their low identity with respect to the sequences present in databases. However, this did not affect the congruence of our results, as the preliminary taxonomic classification assigned by Kaiju was corroborated by Blast analysis and finally supported by family-specific phylogenetic analyses. This is now addressed in the Results and Discussion section.

4) an explanation is required for the differential use of "faeces associated" in the names proposed for the viruses

In databases, the names of viruses belonging to families with a high probability of infecting bats (and not originating from the diet) do not usually include the term "faeces associated". On the contrary, it is usually indicated for other families, such as *Genomoviridae*, *Smacoviridae* and, to a lesser extent, *Circoviridae*. For this reason, we have only added "faeces associated" for viruses belonging to the three families mentioned above.

-A justification for the choice of RNA extraction for DNA viral identification is probably required.

The use of RNA extraction kits is an appropriate procedure when the aim is to analyse DNA only, because they provide an adequate yield and allow the same sample to be used in parallel for the study of the RNA fraction. This is our case and the results obtained from the RNA fraction are being used for the preparation of another manuscript. An explanation of their use is now included in the Methods section.

-for circular genomes, please describe how completeness was assessed.

The aim of this work was to present a description of new eukaryotic DNA viruses with complete or near-complete genomes, as mentioned in the Abstract, but not specifically in the Title, which could be modified if required. To estimate the completeness, CheckV was used, which compares MAVGs with a large database of complete viral genomes, including metagenomes, metatranscriptomes and metaviromes. This is not empirical evidence, so it is possible that a tiny fraction of the genome is not represented in the described MAVGs. For viruses with circular genomes, additional evidence was terminal redundancy, which was detected in 73% of cases and which we have now indicated in an additional column of Supplementary table S5. This description has now been added in Methods and Results and Discussion sections.

-for linear genomes, please describe how the completeness of the sequencing in the ends of the molecule was assessed.

Explanation has been provided in the previous answer.

-maybe the phylogenetic representations could be standardised for all families, for the sake of uniformity (and elegance).

We recognise that the phylogenies present diverse styles, but this is a consequence of the highly variable number of taxa included, which has forced us to adapt the style accordingly. Therefore, we consider that adopting a uniform style for all the representations would compromise the adequate visualisation of some of them and we prefer to keep the figures in their original style.

-please include the ethical statements for wildlife sampling that may be appropriate in the country.

Done.

Minor points:

P4L18: please clarify "Homogenates were centrifuged in two rounds"

Done.

P7L46: I have not checked extensively the correspondence of each reference to the claim reported in the text, but ref 50 is not appropriate here. Please verify all other statements.

We have conducted an extensive check of the bibliography, but no additional errors have been detected.

Response to Reviewer #2

As a reviewer, it's essential to appreciate the diligent efforts of authors in this study.

The study highlights bats as reservoirs for numerous viruses with potential zoonotic transmission. Metagenomic tools have been pivotal in uncovering unprecedented viral diversity. While current research predominantly focuses on RNA viruses, authors addressed this bias by analyzing fecal samples from 189 Spanish bats across 22 species using viral metagenomics targeted to DNA viruses. This approach led to the identification of 50 complete or near-complete viral genomes from various families, potentially including 28 new species. The fact that the collected samples are solely focused on DNA viruses is regrettable, given the richness of the bat collection, yet it is entirely at the discretion of the authors. The publication's focus appears somewhat mixed. If it is intended as a discovery study, the authors should provide more detailed genomic explanations of the newly discovered viruses. Alternatively, if the focus is on viral distribution in bats, the newly identified viruses should be screened within their collection. In the conclusion section, the authors have noted that this paper is not intended as an epidemiological study. Consequently, specific enhancements are needed, especially in the Results section. Authors are asserting the discovery of novel viruses, and to support this claim, they should include viral motifs, not only for specific genomes but also detail the similarity of other genomes to their closest viruses for each individual novel virus.

Page 5

line 29 Please, providing a separate paragraph to explain the construction of the family phylogenetic tree can enhance readability for readers.

Done.

Page 6

Line 2. Could you clarify the statement 'Those viruses identified as having a high probability of infecting bats, and not originating from other sources such as diet, were analyzed by PCR'? It seems unclear how PCR was used to determine if a virus is not foodborne. Perhaps this part of the paragraph should be revised. It appears to imply that PCR was conducted to ascertain whether parvoviruses or polyomaviruses are not transmitted through food. However, I believe the intention might be to indicate that the Smacoviridae and Genomoviridae families were excluded from the analysis panel due to their association with foodborne transmission.

PCR was used to distinguish whether each virus identified was present in a single individual or in several individuals of the corresponding pool, which could help to

understand its distribution in the bat species of origin. In addition, as the reviewer rightly points out, *Smacoviridae* and *Genomoviridae* families were excluded from the PCR analysis because of their likely association with foodborne transmission. This aspect has been better explained in the current version of the manuscript.

Line 6-13 *Did you only examine individual samples from positive pools? If your answer yes, why didn't you screen all individual samples for novel viruses, especially considering their potential impact on geographical distribution? Alternatively, did you observe any similarity between the viral genomes of your novel findings and those found in contigs under 1kb?"*

The primary objective of our study was to provide an overview of the genome-wide diversity of DNA viruses present in Spanish bats, but not to conduct an epidemiological or surveillance study to determine their prevalence. However, as mentioned in the previous answer, we did show the results of PCR analysis of individual samples from each pool where a given virus had been detected, which allowed us to observe that half of the viruses analyzed by PCR were present in more than one individual. This suggested to us that these infections were not exceptional, but could be characterised by a high population prevalence, but we considered that a global search study in our entire study population would not be of major relevance to our results. On the other hand, comparison between the described viral genomes and the set of tens of thousands of partial contigs identified would also not increase the relevance of our message. This might be appropriate in a paper examining in detail the diversity of a particular viral family, but not in the present manuscript, which describes globally the results obtained in seven different viral families.

Line 20 *What criteria guided your selection of 50 out of 1053 nearly complete genomes? I suggest including a supplementary table in the publication containing the details of these 1003 complete genomes, which would enhance the value of the publication. Additionally, you have the option to incorporate a chart illustrating the distribution of viral reads into the paper.*

As indicated at the beginning of the Results and Discussion section, from the total number of complete genomes originally identified, we selected those that were associated with vertebrate viral families and added genomes from two poorly known viral families (i.e. *Smacoviridae* and *Genomoviridae*). As suggested by the reviewer, we have added a supplementary table (Supplementary Table S4) where we describe the general characteristics of the complete genomes identified in our bioinformatics analysis.

During the process of compiling this information, it should be noted that we realised an error originally made in the handling of the data had led us to ignore a number of contigs belonging to a particular pool. Reanalysis of these contigs led us to detect two additional complete circovirus genomes, which have been uploaded to Genbank and included in the current version of the manuscript. Consequently, our results now describe 52 complete genomes selected from an initial set of 1259 contigs defined as complete or near-complete genomes.

Line 31. *None of the data submitted to Genbank or SRA has been released yet, please do not forget to release it upon publication.*

As usual, accessions will be automatically released right after publication.

Line 32. *Kindly include the lengths of all novel viruses in parentheses within the paper. E.g. MAVG3 (7674nt)*

When incorporating this information into the manuscript, we realised that it may be illustrative in some families where few viruses were detected, but results in a very cumbersome text in others with a large number of viruses. Consequently, in order to maintain uniformity throughout the descriptions of the different viral families, and since genome length is easily accessible information in Supplementary Table S5 for interested readers, we have chosen to indicate in the main text only the size range of the MAVGs described in each family.

Page 7

Line 10 "Could you please clarify whether the sequence identity section pertains to the entire genome or is specifically based on the L1 region?"

Done.

Line 25 You are claiming novel papillomavirus because of this could you consider including in the paper the genome similarities of each genome individually to its closest virus?"

A more detailed description of the newly described papillomaviruses has been included in the current version and domain/motif specific data are now presented in the new Supplementary Table S6.

Page 8

Line 16 The section discussing novel viruses, particularly polyomaviruses, requires improvement. Please enhance this section by detailing the conserved domains and viral motifs contained within your novel polyomavirus LT antigen.

The information required by the reviewer has now been included in the new version and added in the new Supplementary Table S7.

Line 23 Please add reference.

Done.

Page 9

Line 4 Please provide a detailed information about your novel parvovirus such as length of NS1, VP, ORFs, motifs etc.

This information has been added in the main text and included in the new Supplementary Table S8.

Response to Reviewer #3

In this study, authors have analyzed DNA viruses, through metagenomics of fecal samples collected from 189 Spanish bats belonging to 22 different species. They report:

(i) Recovery of 50 complete or near-complete viral genomes, which belong to 7 families of DNA viruses.

(ii) 28 of the identified genomes are new viral species.

(iii) The raw data and MAVGs are deposited in Genbank (Sequence Read Archive of GenBank under accession numbers SRR27912327-51 and the complete or nearly complete genomes, under accession numbers PP410048-97).

Strengths of the study:

- 1. Context of the study is presented very well.*
- 2. Relevant tools and methods are used for data analysis.*

Weaknesses of the study:

Gut microbiome in bats includes bacteria and therefore bacteriophages can be expected in addition to the eukaryotic viruses. In this report, authors found viruses that belong to various families of eukaryotic virus families only. It would be good to include this point in the discussion because in other organisms the identified viruses in the gut are mostly bacteriophages.

In our work, we focused exclusively on the characterisation of eukaryotic DNA viruses. Therefore, one of the primary filtering steps in the taxonomic classification of viral contigs was to exclude those assigned as bacteriophages (described in the Methods section). There is no doubt that our sequencing data show a majority fraction of phages, as evidenced by the new supplementary table S4, which includes the listing and tentative taxonomic classification of the more than 1,000 contigs associated with complete or near-complete genomes. We preferred, however, to omit their description in order to keep the focus on our main message. In any case, as suggested by the reviewer, we have added a comment in the Results and Discussion section where we emphasise that the type of samples analysed in this study contain a large number of phage sequences that deserve further study.

Re: Spectrum00675-24R1 (Full genome sequencing of dozens of new DNA viruses found in Spanish bat faeces)

Dear Dr. José M Cuevas:

Thank you for the privilege of reviewing your work. Below you will find my comments, instructions from the Spectrum editorial office, and the reviewer comments.

You have addressed most of the concerns raised by the two reviewers, but there are still some issues you need to consider. Besides, please provide an explanation that an IACUC approval document is not required for this study.

Revision Guidelines

Sincerely,
Biao He
Editor
Microbiology Spectrum

Reviewer #1 (Comments for the Author):

The authors have properly answered most of the comments, but some of the question remain. I detail them here.

Response to point #2 is very elusive. It would be important to discuss why there is so little host genomic DNA. This is most likely related to the depletion in gDNA that is associated to the RNA purification approach used by the authors, and for which they

have taken advantage of co-purification of short DNA fragments with the RNA. Should this be the case, this bias may also underlie the increased likelihood of recovering DNA from small viruses and possibly also from viruses with circular genomes compared to linear ones. Again, this bias merits discussion.

The title clearly states full-length genome. It is essential that the communicated findings adhere to this completeness.

Response to point #4 remains problematic. It is difficult to understand how a genome of the size anticipated for papillomaviruses can display a BLAST hit corresponding to only 1% (MAVG3) or 3% (MAVG3) of the genome, and be identified as such; and the same holds true for MAVG11 and Cycloviruses. The GenBank entries have not been made available yet, and I cannot evaluate the appropriateness of the viral family associations. The fact that Kaiju and Blast yield the same results is not surprising because Kaiju uses blastp, so that this is not an argument in itself as they are not independent from one another.

Also in point 4, a number of algorithms take into account the circular nature of the assembled genomes and should be used to correctly assess genome completeness.

Reviewer #3 (Comments for the Author):

My point has been addressed satisfactorily.

Authors have addressed my point satisfactorily.

Response to the Editor

Thank you for the privilege of reviewing your work. Below you will find my comments, instructions from the Spectrum editorial office, and the reviewer comments.

You have addressed most of the concerns raised by the two reviewers, but there are still some issues you need to consider. Besides, please provide an explanation that an IACUC approval document is not required for this study.

In order to complement the previous explanation provided in the Response-to-reviewers, we provide further details below as to why our study does not require an IACUC authorisation. The key point is that our work only consists of obtaining a faecal sample, which does not involve animal experimentation according to the European Directive 2010/62/EU (Article 1, point 5; Page 276/39; marked in yellow in the file “European-Directive_Animal-Protection”) and transposed literally into Spanish law (Article 2, point 5; Page 11372; marked in yellow in the file “Spanish-Law_Animal-Protection_English”). That said, the execution of the fieldwork only requires a regional authorisation, as attached (“Regional-authority-permit_English” file). For the granting of such regional authorisation, the responsible agency of the regional government evaluates the specific conditions of the requested fieldwork, even if the proposed procedures involve animal experimentation, which is not the case, as mentioned above. As a general rule, the IACUC of our institution (Universitat de València) does not have the competence to authorise a procedure that would be carried out outside the institution's facilities, such as fieldwork, as it could not be subject to its supervision (Personal communication from a member of the IACUC of the Universitat de València; Dr. Enrique Font, enrique.font@uv.es; Professor of Zoology). For these reasons, and to clarify this issue in the body of the manuscript, the “Ethics approvals” section has been modified to accommodate the above explanation (**Page 15, lines 8-16**).

Response to Reviewer #1

The authors have properly answered most of the comments, but some of the question remain. I detail them here.

Response to point #2 is very elusive. It would be important to discuss why there is so little host genomic DNA. This is most likely related to the depletion in gDNA that is associated to the RNA purification approach used by the authors, and for which they have taken advantage of co-purification of short DNA fragments with the RNA. Should this be the case, this bias may also underlie the increased likelihood of recovering DNA from small viruses and possibly also from viruses with circular genomes compared to linear ones. Again, this bias merits discussion.

Honestly, we are confused about the reviewer's point of view on this point. Regarding the proportion of host genomic DNA, we consider this aspect to be irrelevant for the description of our results. However, even if it did, this is a factor that we cannot evaluate, as the genome of most of the bat species we have worked with is unknown. Nor have we analysed the proportion of bacterial DNA, because this would not provide information essential to the aim of our study. We have only provided some information on the detection of bacteriophages (which constitute the predominant viral fraction, Supplementary Table

S4), following a recommendation of reviewer 3, although again we have avoided going into detail to avoid diluting the main message of our work.

Going back to the beginning, we do not understand why the reviewer highlights that there is little host genomic DNA, since firstly, such information has not been provided throughout the manuscript and secondly, it would not be possible to obtain it for most of the samples, as discussed above. We interpret this reviewer comment as stemming from the fact that two of the MAVGs described in Supplementary Table S5 are associated with more than one million sequencing reads (and three others with more than 200,000 reads), which might suggest that the viral fraction was predominant in the corresponding sequencing libraries. This, however, is the exception, as most of the MAVGs described represent a very low percentage of the total reads. In any case, the detection of some highly represented MAVGs is common in viromics studies, as we already stated in the first revised version of the article, where a full paragraph was added on page 7 in response to the reviewer's previous comment in this regard. We have extended the content of this paragraph (**Page 7, lines 14-18**) in the current version in an attempt to be more enlightening, and hopefully this is sufficient to address the reviewer's concern.

Regarding the purification method, we understand that the reviewer assumes that our procedure favours the collection of RNA, but this is not the case. As specified in the manufacturer's manual, "The QIAamp Viral RNA Mini procedure is not designed to separate RNA from cellular DNA". To do so, although this was not our scenario, it would be necessary to use other additional procedures, such as treatment with DNases. This kit, as we indicated in the previous version of the manuscript, and emphasised in the current version by adding new references (**Page 4, lines 23-26**), is commonly used to recover RNA and DNA viruses. For consistency with this, we have changed the title of the corresponding subsection of Material and Methods, which now reads "nucleic acids extraction" instead of "DNA extraction". (**Page 4, line 11**). In our study, beyond considerations on experimental methodology, we have generated the bias ourselves, given that our premise was the description of viruses with complete or near-complete genomes, which obviously leads to the identification of viruses with small genomes, with the exception of adenoviruses. This aspect has now been addressed with an additional comment in the Conclusions section (**Page 14, lines 29-36**), but we do not think it is necessary to extend the discussion.

The title clearly states full-length genome. It is essential that the communicated findings adhere to this completeness.

We consider the title of our manuscript to be consistent with the results presented, although we obviously cannot be absolutely certain that all the genomes presented in our study are complete, given that they have been inferred on the basis of bioinformatics procedures. This is reflected in the summary, which in our view is sufficiently informative. The alternative would be to modify the title of the article by deleting "Full genome", although we believe that this would be detrimental to the information provided to readers.

Response to point #4 remains problematic. It is difficult to understand how a genome of the size anticipated for papillomaviruses can display a BLAST hit corresponding to only 1% (MAVG3) or 3% (MAVG3) of the genome, and be identified as such; and the same

holds true for MAVG11 and Cycloviruses. The GenBank entries have not been made available yet, and I cannot evaluate the appropriateness of the viral family associations. The fact that Kaiju and Blast yield the same results is not surprising because Kaiju uses blastp, so that this is not an argument in itself as they are not independent from one another.

We agree with the reviewer that it is surprising that four of the MAVGs identified have such low query coverage, but we can simply state that this is the result of Blast with default parameters. Our interpretation, as indicated previously, lies in a very low identity with respect to the sequences currently present in databases. In any case, the results obtained with Kaiju and blast, which are not independent, as the reviewer points out, simply helped us to tentatively assign each MAVG to a specific viral family. Subsequently, family-specific phylogenetic analyses corroborated the previous assignment in all cases. In the particular case of the *Papillomaviridae* family, to which two of the four MAVGs with low query coverage (i.e. MAVG3 and MAVG46) were provisionally assigned, all new sequences in our study for this family were submitted to the International Animal Papillomavirus Reference Center, who provided us with the taxonomic classification at species/genus level, as well as the proposed nomenclature (now explicitly stated in the manuscript, **Page 8, lines 37-39**). It is also worth mentioning that all four MAVGs had terminal redundancy and could therefore be considered complete genomes. In addition, a Blastp analysis with the largest ORF for each of them (now added in Supplementary Table S5) showed a query coverage ranging from 76 to 100% and corroborated the initial taxonomic classification. This information has now been provided in an extended explanation (**Page 7, lines 35-44**). Overall, all subsequent results were congruent with the tentative Blast classification, not only with respect to phylogenetic analyses, but also in terms of the structural features of these four MAVGs, including expected genomic sizes/ORFs, and presence of domains, motifs and regulatory elements specific to the different viral families initially assigned by Blast.

Also in point 4, a number of algorithms take into account the circular nature of the assembled genomes and should be used to correctly assess genome completeness.

On this point, we do not understand the reviewer's argument. In our study, we have used CheckV (PMID: 33349699), an automated pipeline for estimating genome completeness, which is, in our opinion, considerably more accurate than alternative approaches. For example, in the article describing this pipeline, a comparison is made with viralComplete (PMID: 32413137), which estimates completeness based on affiliation to viruses from the NCBI RefSeq database, and VIBRANT (PMID: 32522236), which classifies sequences based on gene content and the presence of direct terminal repeats. CheckV showed a better performance in both comparisons. To illustrate this, a commentary on the use of CheckV and its comparison with other procedures has been added in the Results and Discussion section (**Page 7, lines 6-10**). In addition, since its publication in 2021, CheckV has been cited ca. 500 times and, in our view, has become the current gold standard for genomic completeness estimation. Thus, we consider that using other bioinformatics approaches would not result in greater rigour in our results. However, if the reviewer considers it strictly necessary, we would be grateful to know which tools would be more useful, as we have not been able to identify suitable candidates in our search.

Response to Reviewer #3

My point has been addressed satisfactorily.

We are pleased that reviewer 3 considers that its comments have been addressed adequately in the previous version of the manuscript.

Re: Spectrum00675-24R2 (Full genome sequencing of dozens of new DNA viruses found in Spanish bat faeces)

Dear Dr. José M Cuevas:

Your manuscript has been accepted, and I am forwarding it to the ASM production staff for publication. Your paper will first be checked to make sure all elements meet the technical requirements. ASM staff will contact you if anything needs to be revised before copyediting and production can begin. Otherwise, you will be notified when your proofs are ready to be viewed.

Sincerely,
Biao He
Editor
Microbiology Spectrum

Reviewer #1 (Comments for the Author):

It may belong to the scientific construction system that authors and reviewers disagree. I do not want that this disagreement impedes publication of these results and of this work, which is overall of good quality.